# Beyond Pick-and-Place:
# Tackling Robotic Stacking of Diverse Shapes

**Alex X. Lee**[*], **Coline Devin**[*], **Yuxiang Zhou**[*], **Thomas Lampe**[*],
**Konstantinos Bousmalis**[*†], **Jost Tobias Springenberg**[*†],
**Arunkumar Byravan, Abbas Abdolmaleki, Nimrod Gileadi, David Khosid,**
**Claudio Fantacci, Jose Enrique Chen, Akhil Raju, Rae Jeong,**
**Michael Neunert, Antoine Laurens, Stefano Saliceti, Federico Casarini,**
**Martin Riedmiller, Raia Hadsell, Francesco Nori**

DeepMind, London, UK

**Abstract:** We study the problem of robotic stacking with objects of complex geometry. We propose a challenging and diverse set of such objects that was carefully designed to require strategies beyond a simple "pick-and-place" solution. Our method is a reinforcement learning (RL) approach combined with vision-based interactive policy distillation and simulation-to-reality transfer. Our learned policies can efficiently handle multiple object combinations in the real world and exhibit a large variety of stacking skills. In a large experimental study, we investigate what choices matter for learning such general vision-based agents in simulation, and what affects optimal transfer to the real robot. We then leverage data collected by such policies and improve upon them with offline RL. A video and a blog post of our work are provided as supplementary material.[3]

**Keywords:** sim-to-real, offline RL, manipulation, stacking, robot learning

## 1 Introduction

There has been a plethora of work on applying learning algorithms to solving difficult real robot manipulation problems in a general way with a large and diverse set of objects. However, the focus of existing work has primarily been on tasks like grasping [1, 2, 3, 4], which do not typically require complex inter-object contact dynamics. Manipulation tasks which involve interactions between diverse objects, e.g. construction of simple structures, require skills and control strategies that are more complex than and qualitatively different from simply grasping and placing such objects.

As a step in this direction, we study whether, and how, we can tackle stacking of diverse objects in the real world via a learned policy, using only information from RGB cameras and proprioception. Several works have recently considered vision-based real-robot stacking [5, 6, 7, 8, 9, 10, 11]. However, the focus has almost exclusively been on stacking cube-like objects (which does not require reasoning about orientation, shape, or stack stability), or on a limited set of known irregular objects with limited diversity [5]. Additionally, prior work reported low task success rates, pointing towards the difficulty of stacking even cuboids in the real world. In order to study stacking in a principled way, we propose a new benchmark, *RGB-Stacking*, see Figure 1, which consists of a set of 152 procedurally-generated and 3D-printed rigid geometric objects. These were carefully designed to pose different degrees of grasping and stacking difficulty for a parallel gripper on a 4-DoF arm. The benchmark is standardized and we release all relevant information for replicating it as supplementary material.[4] What differentiates our work are two primary characteristics: our large variety of objects and our extensive real world evaluations.

To highlight the challenges in stacking these objects, we instantiated two RGB-stacking tasks (with corresponding versions in simulation and real world) designed to investigate a set of research questions. In the first task, we consider mastering stacking for a set of 5 specific combinations of objects.

---

[*]Equal contribution.

[†]Corresponding authors: konstantinos@deepmind.com and springenberg@deepmind.com.

[3]Video: dpmd.ai/robotics-stacking-YT. Blog post: dpmd.ai/robotics-stacking.

[4]Simulated environment and RGB-objects: https://github.com/deepmind/rgb_stacking.

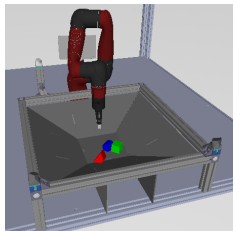 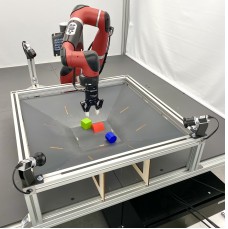 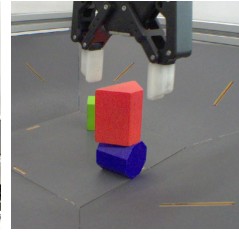 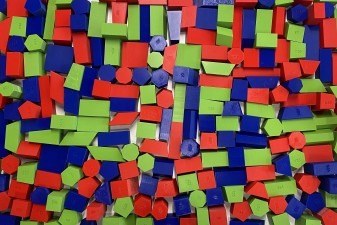

**(a)** Simulation environment    **(b)** Real-robot environment    **(c)** Successful stack example    **(d)** RGB-objects

**Figure 1:** The RGB-stacking tasks involve three objects colored **R**ed, **G**reen, and **B**lue to signal to a vision-based agent which one should be stacked on top of which. We present tasks for stacking red objects on top of blue ones, with the green ones as distractors. The variations in shape require successful agents to reason about contact dynamics and object orientation to form stable stacks. The benchmark defines a parametric family of RGB-objects, which allows designating held-out objects that are quantifiably different from the training objects.

These, shown in Figure 3, were chosen in terms of the challenges they present: precision in grasping and stacking, balancing, and using the top object as a tool to flip a bottom object that has a slanted face pointing up. For this task, we investigated whether it is possible to learn a single vision-based policy that can reliably stack all 5 combinations in the real world. For the second task, we considered the challenge of learning general stacking strategies from a large set of training objects and test how they transfer to held-out objects—to assess generalization. In both cases, we find that using an agent trained in simulation is the most efficient way to bootstrap data collection for offline RL.

Our contributions are as follows. *(a)* We present and release a benchmark for stacking that features a diverse set of geometric objects and tens of thousands of possible stacks. *(b)* We show that it is possible to learn a vision-based policy that can stack multiple combinations of objects and can demonstrate a variety of stacking strategies for non-cuboid objects, emergent from RL training. *(c)* We present the framework we used to obtain our results without the need for human demonstrations. Specifically, we train vision-based agents with domain randomization via an interactive distillation approach—decoupling learning the required stacking skills with RL training from mapping the skills to perception with imitation. We also ablate individual components of this pipeline, requiring more than 54 000 stacking attempts on real robots. *(d)* Finally, we show that an offline RL step using data gathered on the real robots can boost performance when the data is collected with sim-to-real policies; but doing so using real episodes from a scripted agent was worse than zero-shot sim-to-real.

## 2 Related Work

**Robotic Stacking.** Several works have recently dealt with vision-based real-robot stacking either by learning a curriculum of the different stages of the task [6], by combining human demonstrations and RL [7, 8], or by using some flavor of sim-to-real transfer [7, 9, 10]. In most prior work, the focus has almost exclusively been on cube stacking. Furrer et al [5] deals with stacking 6 known stones of different shapes. These stones however all have fairly wide support, high friction, require similar pick-and-place strategies, and only 11 episodes were used for evaluation of the suggested method. In contrast, RGB-stacking provides a general, reproducible, and significantly more difficult benchmark for robotic manipulation. It involves thousands of stacking combinations and our evaluations are significantly more extensive than in prior work.

**Large-Scale Deep RL for Robot Manipulation.** Deep learning for robotic manipulation was in part popularized by large-scale data collection for grasping in the real world [1, 2, 3, 4], a task that allows for automated evaluation and resetting. However, these efforts considered problems with short time horizons and limited dynamical effects (e.g. position-controlled robots with actions taking up to a second). As a result they often used simple Q-learning from collected data with direct optimization for action selection [3]. In contrast we consider velocity-based control at a rate of 20 Hz leading to long episodes (400 timesteps). This is a more challenging scenario in which learning with, e.g., QT-Opt [3] from recorded data alone would exhibit problems due to wrongly-estimated Q-values [12]. Additionally, large-scale RL-based learning in the real world can be difficult to reproduce as the entire real-world process (which can last months [2, 3]) is a function of the initial conditions. In contrast, our simulation-based training is inherently reproducible. In-hand object manipulation provides an alternative route for a difficult challenge, but automating episodic resets become more challenging as the objects easily fall out of the hand [13, 14].

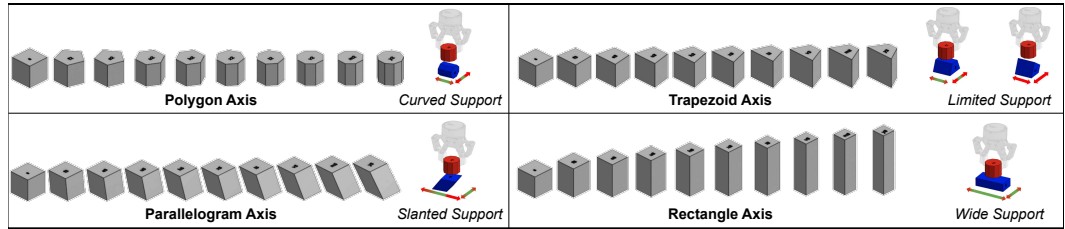

**Figure 2:** Illustration of the deformations applied for each of the 4 major axes of the RGB-Objects parametric family. Each deformation changes the stacking affordance of RGB-objects; for bottom objects the stacking support is illustrated with red-green lines showing where a stack is possible (green).

**Imitation Learning and (Offline) Reinforcement Learning.** For training in simulation, and subsequent training of sim-to-real policies, we use techniques from off-policy RL followed by an interactive imitation learning approach. For training in simulation we consider RL with full state-information, in order to avoid problems with partial observability. In this setting we use the MPO algorithm [15]; a sample efficient, state-of-the-art, off-policy actor-critic method. Initial training is then followed by our pipeline for obtaining a (general) vision policy via a DAgger [16] style interactive imitation learning approach to distill the MPO policy. We found this to significantly outperform Behavior Cloning [17] / policy distillation [18] due to the fact that DAgger-style training provides corrections for mistakes of the vision-based policy. We use Domain Randomization [19, 20, 21, 22, 23, 24] to obtain good sim-to-real transfer in the absence of real-world data; more details on this are given in the supplementary. Finally, when improving policies with offline RL from real data we consider: i) a filtered behavior cloning loss that is equivalent to CRR-exp [25] but with data from a single policy source (analogous to work on combined offline and online RL [26, 27]); ii) Behavior Cloning based on only successful trajectories. Both allow for stable offline learning without the problems standard RL algorithms exhibit in this setting [12].

## 3 The RGB-Stacking Benchmark

Despite stacking being addressed as a robotic learning task in prior work, previous approaches have been limited to a reduced set of objects, typically cubical in shape. We focus on the problem of stacking a variety of objects characterized by different shapes. This is obtained by defining a parametric family of *RGB-objects*. The design principle is to vary the grasp and stack affordances of these objects for a parallel gripper. Our choices significantly change the difficulty of the stacking task by requiring an agent to exhibit behaviors that go beyond a simple pick-and-place [28] strategy. We do so while keeping the benefits of automated learning and evaluation as in [29, 2]. In Appendix A, we analyze the difficulty of the task in various ways. We qualitatively evaluate the grasping affordance based on general principles on force closure and object funnels [30], and quantitatively based on the Ferrari-Canny grasp metric [31, 32]. Similarly, we qualitatively depict the stacking affordance (Figure 2), which is varied by having bottom objects, whose top flat surfaces differ in area, shape, and orientation. We also quantitatively evaluate both affordances by evaluating the stacking performance of human teleoperators in simulation, and of a carefully scripted agent.

### 3.1 The RGB-Objects Family

Our objects are all obtained by applying a deformation to a *seed* cube, a 2D vertical extrusion of a planar square. We defined 4 major axes of deformation, illustrated in Figure 2, resulting in different shapes. These shapes can be thought of as the vertical extrusion of a 2D shape. The **Polygon Axis** transforms the planar square into a regular polygon. The **Trapezoid Axis** morphs the planar square to an isosceles trapezoid. The **Parallelogram Axis** changes the orientation of the extrusion axis, from vertical to progressively more slanted axes. Lastly, the **Rectangle Axis** uniformly scales the object along the x, y or z-axis.

These deformations and their combinations form a parametric family of objects. We obtain the final set of objects by uniformly sampling, *for both the major axes and all pairwise combinations*, 8 objects between the seed object and a maximally deformed object. Figure 1(d) shows a subset of these in the real world, and Appendix A contains a complete depiction.

## 3.2 RGB-Stacking Tasks

The RGB-stacking tasks we tackle in this paper involve three objects in a basket in front of the robot. Those are colored **R**ed, **G**reen, and **B**lue to signal to a vision-based agent which one should be stacked on top of which, and which one is just a distractor. In all our experiments, red is assigned to the top object, blue to the bottom object, and green to the distractor. We note that the role of the latter is not just to distract visually but also to serve as an obstacle during stacking. A successful stack is achieved when the red object is above the blue one and their centroids are vertically aligned within some thresholds. A detailed description can be found in Appendix B.

### 3.2.1 Skill Mastery on 5 Specific Triplets

The first task involves 5 RGB-object triplets, with each object in a triplet being from a *major* axis. The triplets were chosen to have different degrees of difficulty for a stacking agent. The task is to achieve skill mastery on these 5 fixed combinations, shown in Figure 3, with a single vision-based agent. We explain the challenges for each triplet in a loosely descending order of difficulty:

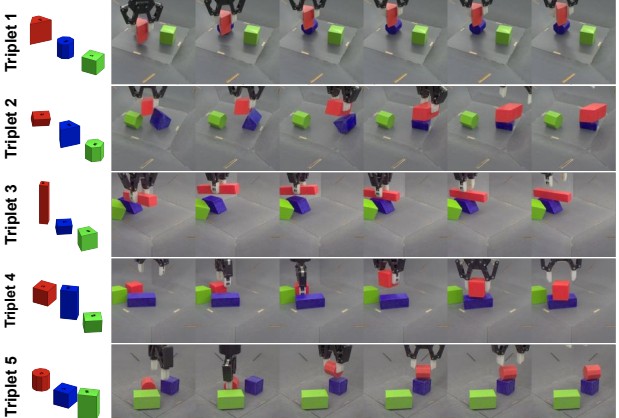

**Triplet 1.** The main challenge is grasping the top object, a grasp that involves the gripper closing on the slanted sides will fail. The bottom object also provides difficulty, as it has a limited stacking surface and can easily roll.

**Triplet 2.** In this triplet, the bottom object has sloped sides and may need to be reoriented by using the the top object as a tool before stacking.

**Triplet 3.** The challenge with this triplet is to have a secure central grasp for the elongated object and balance it on top of the slanted bottom object.

**Triplet 4.** An easy triplet, it has rectangular prisms for both red and blue objects; the challenge is primarily to align their centroids, required for a stack to be considered successful.

**Triplet 5.** In this triplet, the top object can easily roll off the bottom object as it has a large number (10) of faces and is nearly cylindrical.

**Figure 3: Stacking Skill Mastery for our 5 Specific Triplets**. All the 5 stacks shown here are performed by a single agent trained solely in simulation with domain randomization and transferred to the real robot in a zero-shot fashion. Each triplet poses its own unique challenges to the agent: Triplet 1 requires a precise grasp of the top object; Triplet 2 often requires the top object to be used as a tool to flip the bottom object before stacking; Triplet 3 requires balancing; Triplet 4 requires precision stacking (the object centroids need to align); and the top object of Triplet 5 can easily roll off if not stacked gently.

### 3.2.2 Skill Generalization

The second RGB-stacking task we are tackling is a transfer task. For this we designate the axes of all pairwise combined deformations (see Appendix A for details) as the *training axes* and the ones of single deformation—the major axes above—as the *held-out axes*. Based on these, we created a *training object set*, which consists of 103 different shapes, and a *held-out set* (containing 4 axes of deformation), containing 40 shapes. The 5 fixed triplets chosen for the previous task belong to the held-out axes and final performance is evaluated based on them.

## 4 Stacking a Diverse Set of Objects in the Real World with a General Policy

Since we have access to a simulator with a reasonably accurate model of our real robot and its cell, our approach is to solve the RGB-stacking tasks in three stages: 1) RL training of an expert in simulation using state information, 2) imitation of such an RL expert with a vision-based policy in simulation with domain randomization for both visual appearance and dynamics, and 3) a policy improvement step using data gathered on the real robots. Explicitly decoupling these steps allows for faster and cheaper iteration by enabling parallel experimentation and tuning for all three stages.

**Notation.** We consider the Markov Decision Process (MDP) in which a policy $\pi(a|s)$—a probability distribution over actions $a \in \mathcal{A}$ that is conditioned on states $s \in \mathcal{S}$—acts to maximize the discounted sum of rewards, with sparse rewards $r(s, a)$ and discount factor $\gamma \in (0, 1]$. We use $Q^\pi(s, a) = \mathbb{E}_{\rho_\pi}[\sum_{t=0}^\infty \gamma^t r(s_t, a_t)|s_0 = s, a_0 = a]$ to denote the Q-function, where $\rho_\pi$ is the trajectory distribution induced by $\pi$, and $A^\pi(s, a) = Q^\pi(s, a) - \mathbb{E}_{a' \sim \pi}[Q(s, a')]$ for the advantage when executing $\pi$. In the real world, we cannot assume access to full state information (e.g. object poses and labels), so both the policy and advantage are defined based on observations $o(s) \in \mathcal{O}$ (actuator positions of the robot and two camera images) as $\pi(a|o(s))$ and $A^\pi(o(s), a)$, respectively.

## 4.1 Training Expert Policies from State Features in Simulation

We train expert policies via off-policy RL in simulation with a shaped reward. We used the MPO algorithm [15] for this purpose (details in Appendix D), but any RL algorithm could have been used in this stage. We found training directly from state $s$ to be significantly faster than training from vision and thus trained expert policies $\pi_e(a|s, y)$—where $y$ denotes a triplet number in $\{1, \ldots, 5\}$ for the *Skill Mastery* task, or the parameters of object deformation for the *Skill Generalization* task.

## 4.2 Interactive Imitation Learning of a Vision-Based Policy for Sim-to-Real Transfer

We use imitation in simulation to distill the state-based expert(s) into a single "student" vision-based policy $\pi_\theta^{\text{vis}}(a|o(s))$ that can be executed in the real world, where accurate object state is unavailable, and that is designed for maximizing sim-to-real transfer using components described in Section 5.2. We could attempt to train this policy via supervised Behavior Cloning (BC) [17] on a dataset of trajectories from the teacher, but we find this suffers from covariate shift, where the student visits different states than the teacher (in which it fails). We instead perform DAgger-style [16] interactive imitation learning (IIL) of teacher actions on states from trajectories obtained by executing the *student* policy. We refer to this as IIL, in which we perform optimization alongside data-collection (rather than alternating collection and supervised learning as in [16]). Specifically, our student policy $\pi_\theta^{\text{vis}}$ is continuously executed in a *heavily domain randomized simulation environment* (see appendix for details), collecting trajectories stored in a replay buffer $\mathcal{D}_{\text{vis}}$. The student is trained to maximize:

$$\mathcal{L}_{\text{IIL-s2r}}(\theta) = \underset{(y, \tau) \sim \mathcal{D}_{\text{vis}}}{\mathbb{E}} \left[ \sum_{s_t \in \tau} \log \pi_\theta^{\text{vis}}(a_t|o(s_t)) \big| a_t \sim \pi_e(\cdot|s_t, y) \right], \tag{1}$$

where $(y, \tau)$ is a pair of object information $y$ and trajectory $\tau = (s_0, a_0, s_1, \ldots, s_T)$. Note that if we replace the collected data $\mathcal{D}_{\text{vis}}$ with expert data $\mathcal{D}_e$ from executing the teacher $\pi_e$, then we recover standard BC but with infinite data. The form of the teacher and the data above depends on the task.

**Skill Mastery.** We have access to 5 teachers indexed by the object triplet number $y \in \{1, \ldots, 5\}$. Each teacher $\{\pi_e(a|s, y = 1), \ldots, \pi_e(a|s, y = 5)\}$ is independently trained only on the respective triplet $y$. The student collects trajectories on all triplets, chosen at random for every episode.

**Skill Generalization.** We use the same technique for the generalization task, except that now we have access to a single general teacher $\pi_e(\cdot|s, y)$. This generalist is conditioned on parameters $y$ of object deformation (see Figure 2) and can specialize to different objects. During IIL, trajectories in $\mathcal{D}_{\text{vis}}$ are from the training objects, as the test objects will not be seen until final evaluation.

## 4.3 One-Step Offline Policy Improvement for Sim-to-Real Policies

Once we obtained a sim-to-real policy, we consider improving its behavior on the real robots. We perform a single step of policy improvement in which we: 1) deploy $\pi_\theta^{\text{vis}}$ on the robots to collect a dataset $\mathcal{D}_{\text{real}}$, 2) train a new policy $\pi_\theta^{\text{imp}}$ offline, using a filtering function $f(s, a, \tau)$, and 3) deploy the improved policy on the robot again for evaluation (1-3 could be iterated to obtain a policy improvement loop). The offline training step aims to maximize the following objective:

$$\mathcal{L}_{\text{IMP}}(\theta; f) = \underset{(y, \tau) \sim \mathcal{D}_{\text{real}}}{\mathbb{E}} \left[ \sum_{s_t, a_t \in \tau} f(s_t, a_t, \tau) \log \pi_\theta^{\text{imp}}(a_t|o(s_t)) \right]. \tag{2}$$

We use the filter $f(s, a, \tau) = \exp(A^{\pi_\theta^{\text{imp}}}(o(s), a)/\alpha)$ with temperature $\alpha$ (a hyperparameter) and learned Q-function (to calculate advantage), referred to as CRR-IMP, which recovers the CRR-exp [25]/AWAC [26] objectives, but with data from the deployed policy only. Unlike in Section 4.1, *we here use a binary, sparse, reward* for success per step. We also consider $f(s, a, \tau) = r(s_T)$, referred to as BC-IMP, which corresponds to BC on successful trajectories only (stack in last frame).

# 5 Experiments: Solving RGB-Stacking

We outline our efforts to solve the two stacking tasks we proposed in Section 3.2. In both tasks we evaluate our agents on 5 fixed object triplets, and we assume access to unlimited simulated data but only to a fixed budget ($< 100\,000$) of real episodes. In *Skill Mastery* we have access to the fixed object triplets during all stages of training. In *Skill Generalization* our agents only have access to the training object set; the 5 fixed triplets are now held out both in simulation and the real world.

In all experiments we use a standardized evaluation protocol: the positions of the objects are randomized at the beginning of each episode, and the agent has 20 seconds at $20\,\text{Hz}$ to stack the red object on the blue one. We define task success as having a stack at the *end* of the episode. This definition was chosen to exclude cases in which a stack is briefly achieved incidentally in the middle of an episode when the red object is above the blue one but subsequently falls off. In simulation, we evaluate on the training object set by running each policy on 5000 random object triplets for 2 episodes each. For the 5 fixed test triplets, we evaluate each policy for 1000 episodes per triplet. In the real world, we only evaluate on the test triplets, as evaluating on a sufficiently large sample of training objects to obtain statistics is impractical. We use one robot per triplet (evaluating for 200 episodes), for a total of 1000 episodes per policy. Mean performance is calculated by averaging the success rate across the triplets. For state-based policies and vision-based policies trained on real data we report the average across 2 training runs (i.e. random seeds), and for the vision-based policies distilled from a teacher we report the average across 4 runs: 2 distillation runs for each of the 2 teacher seeds. Results for all runs individually are in Appendix E.

## 5.1 Evaluation on the Simulation Stacking Benchmark

We first evaluate several baselines on our simulated benchmark task. We are interested in the following questions: *(a)* How difficult is the task in simulation? *(b)* Would it be preferable to train directly from vision? *(c)* How does interactive imitation and the design choices for it affect performance compared to the state-based teacher?

To assess the task difficulty, we tuned a scripted agent with access to the object positions and evaluated it on both our training set and our test triplets. We also hired 4 individuals with no relation to the research team to attempt the sim task via teleoperation and a game pad for 846 episodes. For details on these indicative baselines, see Appendix C. These demonstrate the difficulty of the task as in both cases the success was lower than 50% (see Table 1).

Next, we evaluate our benchmark tasks in the dense reward setting with MPO from state. A first finding was that the state agents, which are given the full pose of each of the three objects, learn the task 10 times faster than the vision agent; as illustrated in Appendix D, using a vision agent to train from scratch on the robot would be impractical for these tasks. The MPO state-based agents obtain 79.3% on the test triples in the skill mastery setting and 68.8% on the training objects in the skill generalization setting, significantly outperforming the scripted agent. However, *generalization* from training to test objects is only slightly higher than the scripted agent's performance (47.8%). This is because these state-based agents are conditioned on object parameters, which are out of distribution for the test triplets. After obtaining vision agents via IIL-s2r we observe (Table 1, bottom) that although the training set performance drops, performance improves for the test triplets, as the agents can now generalize from visual cues to make inferences about the shape.

We next compare the interactive imitation learning setting (IIL) to learning from teacher trajectories with a behavior cloning loss (BC). Rather than explicitly constructing a fixed size dataset, we generate the data on-the-fly by executing the teacher continuously during training. As shown in Table 1, learning from data generated by the student (interactive imitation) is crucial: IIL performed significantly better than the alternative in both task settings.

## 5.2 Evaluation on the Real-World Stacking Benchmark

We then investigate the following questions related to RGB-stacking in the real world: *(a)* Is it possible to solve the challenging tasks on real robots with a single vision policy, and is that better than a scripted baseline tuned for the test triplets? *(b)* How well does zero-shot transfer of sim-trained agents solve the tasks, and which of our decisions were important for zero-shot performance? *(c)* Do we get a single-step improvement with BC-IMP and CRR-IMP from using data collected in the real world? *(d)* And how does the data distribution affect the performance of these algorithms?

| Method | Simulation Success | |
| --- | --- | --- |
| | Train Set | Test Triplets |
| Human teleoperator | - | 46.6% |
| Scripted agent | 45.0% | 43.1% |
| *Skill Mastery* | | |
|    State teacher | N/A | 79.3% |
|    BC | N/A | 52.4% |
|    IIL-s2r | N/A | 74.2% |
| *Skill Generalization* | | |
|    State teacher | 68.8% | 47.8% |
|    BC | 49.4% | 41.2% |
|    IIL-s2r | 64.7% | 56.0% |

**Table 1: Simulation Success.** Comparison of distillation methods for solving, in simulation, our RGB-stacking tasks from vision and proprioception. These methods assume we have either a large dataset generated by a teacher policy (BC) or the teacher policy itself to supervise the student's interaction (IIL). Interactive learning leads to much higher performance than learning only from the teacher's behavior.

| Method | Sim Success | Real Success |
| --- | --- | --- |
| | Test Triplets | Test Triplets |
| *Skill Mastery* | | |
|    IIL-s2r (deterministic) | 71.7% | 67.9% |
|    IIL-s2r (stochastic) | 74.2% | 67.4% |
|    No Transformer | 73.3% | 69.3% |
|    No image augmentation | 72.8% | 65.7% |
|    No action delay | 73.5% | 68.7% |
|    No binary gripper | 66.1% | 20.7% |
|    MSE & no binary gripper | 65.7% | 25.7% |
| *Skill Generalization* | | |
|    IIL-s2r | 56.0% | 51.9% |
|    No Transformer | 51.3% | 45.4% |
|    No object parameters | 53.7% | 41.1% |

**Table 2: Sim-to-Real Transfer Success.** Ablations of the components of the sim-to-real policy. The simulation and real-world evaluations use $20\,000$ and $4000$ episodes for the triplet average, respectively. The choice of using binary gripper actions for the policy seemed to be the most important factor for sim-to-real transfer success, followed by the choice of executing the deterministic policy on the robots.

As described in Section 4.2, we distill a vision-based policy from a single or multiple state-based experts in simulation. These are 5 specialists for the *Skill Mastery* task and a single generalist for the *Skill Generalization* task. We can then directly execute this distilled vision policy on the real robots. As discussed previously, this decoupling allowed us to quickly iterate and investigate what aids sim-to-real transfer for these tasks. We compare, in Table 2, the simulation performance and the zero-shot sim-to-real transfer performance when ablating these various choices for our method. Policies were trained up to 1 million learner steps, and the sim-to-real policy was selected to be the one with the highest performance in the training setting for each task. As our policies are stochastic, for inference we have the option to execute a random sample from the action distribution or its mode (i.e. the action with highest probability). We execute the **stochastic** policy in simulation and the **deterministic** mode of the policy on the real robots, unless otherwise specified. We are able to achieve high performance of 67.9% on the Skill Mastery task and 51.9% on the Skill Generalization task in the real world. Qualitatively, we also see in Figure 3 that we can, with a single vision-based agent, show a diverse set of skills to address the challenges needed to solve the task for the 5 specific triplets we have chosen. Videos of the behavior are in the supplementary.

**Ablations.** We ablate different algorithmic choices in Table 2 (extended table in the supplementary). One of our decisions was on the BC loss for distillation, which can either be the **negative log-likelihood** or the **mean squared error (MSE)**, which respectively correspond to $-\log \pi_\theta^{\text{vis}}(a_e|o(s_t))$ or $\mathbb{E}_{a\sim\pi_\theta^{\text{vis}}}[||a - a_e||^2]$, where $a_e$ are actions from the teacher (the two are not equivalent even for the Gaussian case as the policy variance is state dependent). We also decided to use a hybrid action space, where the 3D Cartesian and angular velocities are modeled as continuous actions (Gaussian), but we have a **binary gripper action** (Bernoulli). This applies to both state-based and vision-based policies. This choice seemed to be the most important factor for sim-to-real transfer success. However, note that the effects are only visible when transferring (corresponding simulation performance does not change as much). For the *Skill Generalization* task, the state-based expert is conditioned on the **object parameters** $y$. However, the object parameters are not passed to $\pi_\theta^{\text{vis}}$—the agent needs to infer physical object properties from image observations. We also chose to use a **Transformer** model [33] after the ResNet vision encoder for the policy's network architecture, which gives the agent access to temporal information with an attention mechanism. The Transformer does not have a significant effect on *Skill Mastery*. However, it seems to be able to utilize the skill variety of a state-based expert conditioned on the object parameters a more successfully, when compared to an MLP model with the same number of weights. This is clear both in simulation and the real world (and we hence used it in all experiments). Random **action execution delays** of $0, 1$ or $2$ timesteps, and standard **image augmentation** (random color perturbations and image translations) did not seem to affect transfer but we decided to include them in our training procedure to increase our agents' robustness to natural perturbations.

| Method | Real-Robot Success | | | | | |
| --- | --- | --- | --- | --- | --- | --- |
| | Triplet Avg. | Triplet 1 | Triplet 2 | Triplet 3 | Triplet 4 | Triplet 5 |
| Scripted agent | 51.2% | 36.3% | 23.0% | 34.4% | 84.9% | 77.6% |
| *Skill Mastery* | | | | | | |
|    BC (scripted agent data) | 53.4% | 34.5% | 26.8% | 38.3% | 84.5% | 82.8% |
|    CRR (scripted agent data) | 43.4% | 20.5% | 28.0% | 38.3% | 64.0% | 66.3% |
|    IIL-s2r (sim-to-real) | 67.9% | 72.8% | 48.8% | 61.0% | 85.1% | 71.9% |
|    —Sim-to-real agent for test triplets data | 69.6% | 76.4% | 52.7% | 60.4% | 86.5% | 72.0% |
|    BC-IMP (sim-to-real agent data) | 74.6% | 75.5% | 60.8% | 70.8% | 87.8% | 78.3% |
|    CRR-IMP (sim-to-real agent data) | 81.6% | 87.3% | 68.3% | 75.3% | 88.3% | 88.8% |
| *Skill Generalization* | | | | | | |
|    IIL-s2r (sim-to-real) | 51.9% | 25.0% | 39.6% | 36.5% | 91.1% | 67.0% |
|    —Suboptimal agent for training set data | 32.6% | 21.5% | 16.5% | 17.0% | 60.0% | 48.0% |
|    BC-IMP (suboptimal agent data) | 49.0% | 23.0% | 39.3% | 39.3% | 77.5% | 66.0% |
|    CRR-IMP (suboptimal agent data) | 55.6% | 31.3% | 41.3% | 42.0% | 81.5% | 81.8% |

**Table 3: Real-Robot Success.** Different approaches for solving our RGB-stacking tasks in the real world. The kind of data used matters significantly for BC and CRR: using scripted agent data does not result in improved performance, but using data from a sim-to-real agent does. Note that the sim-to-real data used were collected by a single agent per task. For Skill Generalization, as data collection with multiple triplets from the training set was exceptionally time-consuming, we used a suboptimal agent trained earlier in our investigation. This agent was not trained with the best settings we now have for sim-to-real. See text for more details.

Our sim-to-real approach also outperforms a scripted agent tuned on the test triplets, as shown on Table 3. For the Skill Mastery task, we further improve results via offline RL with BC (BC-IMP) on 32 651 successful real episodes collected by the scripted agent and with CRR (CRR-IMP) on a total of 67 446 episodes, which included the unsuccessful ones collected by the same agent. Hyperparameters and architecture choices were selected after such an investigation in simulation. As evident in the table, learning from scripted agent data is not better than sim-to-real. Qualitatively, the policies learned seem to exhibit the same "robotic" movements as the scripted policy that generated the data they were trained on. However, a single policy improvement step based on data collected by a sim-to-real agent with an average success of 69.6%: 85 213 episodes (58 979 successful) yields 74.6% (BC-IMP) and 81.6% (CRR-IMP), resulting in policies with remarkable stacking consistency.

As data collection for the training object set is particularly time-consuming, we only did a single collection (38 446 episodes, 37.4% of which were successful) with an earlier iteration of a sim-to-real agent for the Skill Generalization task (performance of 32.6% when evaluated on the test triplets). As before we trained BC-IMP and CRR-IMP on this data and again observed improved performance on the test triplets, this time using only the episodes collected from the training object set. Note that even though the suboptimal sim-to-real agent trained on the training set of objects was performing worse than the scripted agent, and the data in this case was more limited and of different objects, BC-IMP and CRR-IMP trained on this more diverse data lead to a significant improvement. Finally, Table 3 showcases the variation in difficulty of the 5 triplets: all methods perform best on triplets 4 and 5, as these can sometimes be solved with a stereotypical "pick and place" behavior. In contrast, triplets 1, 2, and 3 each require specialized, advanced, strategies as indicated by the large gap between sim-to-real performance and scripted performance in the Skill Mastery task.

## 6 Conclusion

We studied the problem of vision-based stacking with a large variety of geometric objects that require a diverse set of stacking strategies. We propose a benchmark for studying this problem in a principled way, and make significant progress in solving two tasks involving both skill mastery on specific objects and generalization across them. We use simulation-trained agents to collect data in the real world, which in turn can be used for further performance improvement with offline RL. Our best agent is a single vision-based policy that is capable of a variety of stacking strategies and achieves high performance. Finally, we provide thorough real-world evaluations and discuss what is important for solving our tasks. Despite this success, our benchmark still poses many open challenges as, for example, shown by the gap between Skill Mastery and Skill Generalization results and we hope that it can help development of new methods for learning general policies (e.g. by further adaptation at robot deployment time).

## Acknowledgments

The authors would like to thank Christopher Schuster, Nathan Batchelor, Serkan Cabi, Dave Barker, Jean-Baptiste Regli, Yusuf Aytar, Dushyant Rao and many others of the DeepMind team for their support and feedback during our project and the preparation of this manuscript.

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

# Appendix

## Table of Contents

## A  The RGB-Stacking Benchmark

In this section, we provide more details on our proposed benchmark and analyze its difficulty in various ways. We qualitatively evaluate the grasping affordance based on general principles on force closure and object funnels [30], and quantitatively based on the Ferrari-Canny grasp metric [31, 32]. Similarly, we qualitatively depict the stacking affordance (Figure S5), which is varied by having bottom objects, whose top flat surfaces differ in area, shape, and orientation. We also quantitatively evaluate both affordances by evaluating the stacking performance of human teleoperators in simulation, and of a carefully scripted agent.

### A.1  The RGB-objects family

The RGB-objects are all obtained by applying a certain deformation to a cube, which is our *seed object*. We have defined four major axes of deformation (see Figure 2) of the seed object, which

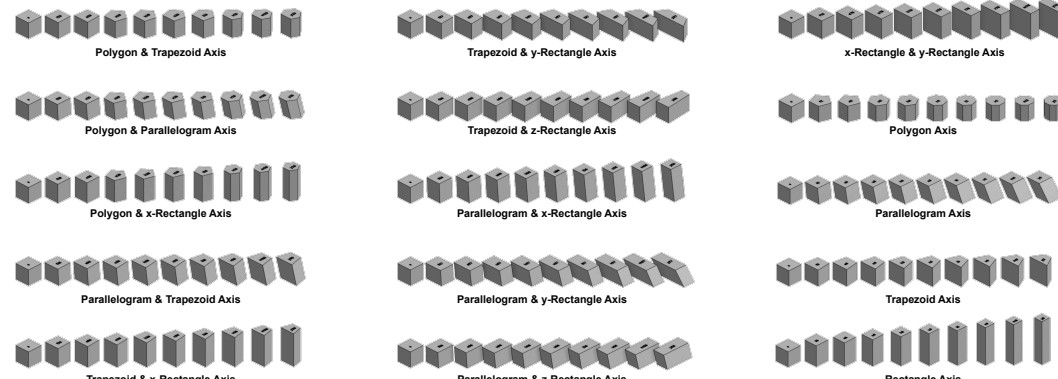

**Figure S1:** Illustration of the deformations applied for each of the 15 axes of the RGB-Objects parametric family (major axes and their unique pairwise combinations). Each deformation changes the stacking affordance of RGB-objects.

result in different shapes. These shapes can also be thought of as the vertical extrusion of a 2D shape, which is also the name of each axis. As already described in the main paper but reproduced here for completion, these are:

- **Polygon-Axis [ℕ]:** deformation obtained by transforming the extruded planar shape (i.e. the square) into a regular polygon.

- **Trapezoid-Axis [$\mathbb{R}_+$]:** deformation obtained by progressively morphing the planar square to a isosceles trapezoid.

- **Parallelogram-Axis [$\mathbb{R}_+$]:** deformation obtained by changing the orientation of the extrusion axis, from vertical (i.e. orthogonal to the plane of the planar shape) to progressively more slanted axes.

- **Rectangle-Axis [$\mathbb{R}_+^3$]:** deformation by uniformly scaling the object along the x, y or z-axis.

These deformations and their combinations define a parametric family of objects. Figure S1 shows a representative sampling of this family. For our *Skill Generalization* task we designate the axes of all pairwise combined deformations as the *training axes* and the ones of single deformation— the major axes above—as the *held-out axes*. The training axes are Polygon & Trapezoid, Polygon & Parallelogram, Polygon & x-Rectangle, Trapezoid & Parallelogram, Trapezoid & x-Rectangle, Trapezoid & y-Rectangle, Trapezoid & z-Rectangle, Parallelogram & x-Rectangle, Parallelogram & y-Rectangle, Parallelogram & z-Rectangle, and x-Rectangle & y-Rectangle. Pairwise mixing of the major axes leads to objects that are duplicates and therefore we omitted certain axes and objects (e.g. the x-Rectangle & y-Rectangle and x-Rectangle & z-Rectangle axes). Also note that the x-, y-, z- Rectangle axes are the same so we refer to these as a single major Rectangle axis.

Based on these 15 axes illustrated in Figure S1, we created a *training object set*, which consists of 103 different shapes, and a *held-out set*, containing 40 shapes. Figure S2 shows a depiction of all the objects that are included in the benchmark and were used in our experiments. The 5 specific triplets chosen for the *Skill Mastery* task belong to the held-out axes with each object in a triplet being the seed or from a different axis[5]. While final performance is evaluated on these 5 fixed test triplets for both tasks, during training for *Skill Generalization* we hold out not just these objects but the entire 4 axes of deformation they belong to. That is, during training we can use the 103 objects from the training object set, while performance is evaluated still on the 5 specific triplet combinations from the *Skill Mastery* task.

---

[5]Technically and for legacy reasons, although the objects for the 5 triplets are sampled from the held-out axes, not all of them are actually depicted in the held-out object set illustrated in Figure S2. As seen in their figure in the main text, 3 are the seed cube object (itself held out), and 4 out of the 15 are actually in the held-out object set. These are the 4 top objects that are not the seed cube. The rest 8 objects are almost identical to existing objects in the held-out object set shown in Figure S2. In the released set of object models, these will be in their own sub-directory.

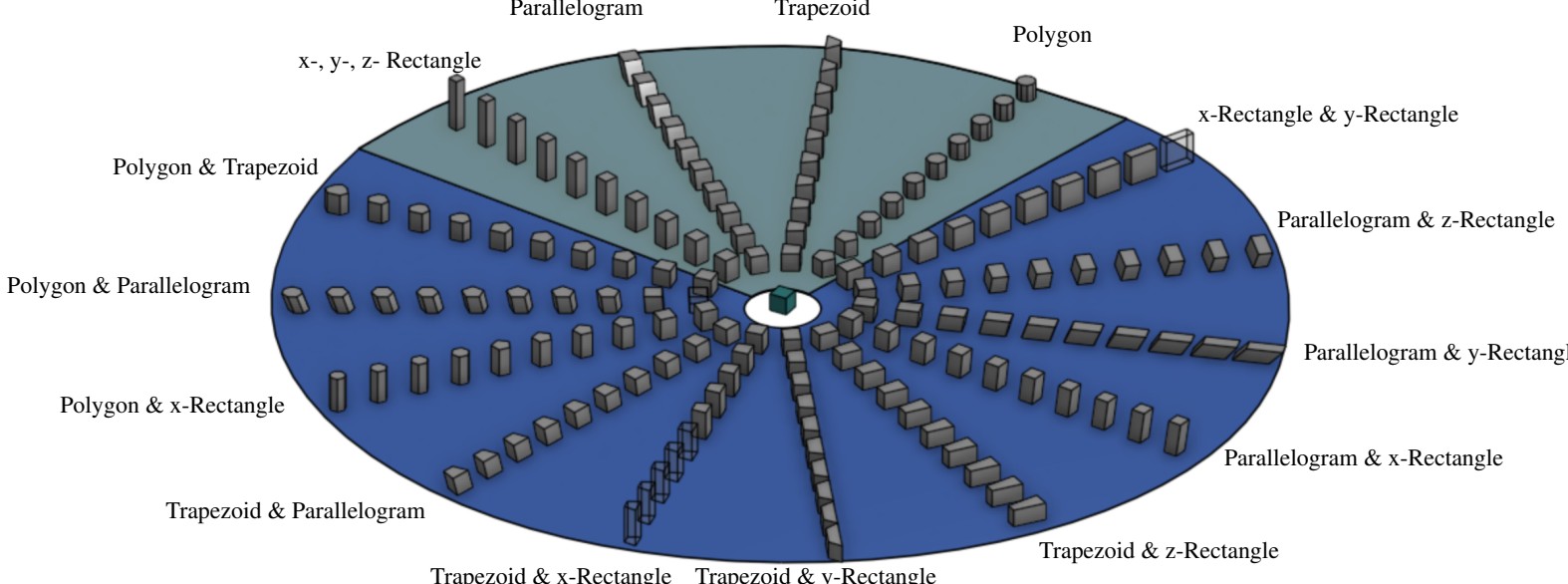

**Figure S2: The RGB-objects that are included in the benchmark grouped according to each of the 15 chosen axes of deformation.** The seed object is at the center; all the other objects are the result of deformations of this cube. These deformations change the grasping and stacking affordances of the objects. The held-out objects (major axes) are enclosed in the teal sector; the training objects (pairwise mixing of two major axes) are enclosed in the blue sector. Some objects cannot be grasped with a parallel gripper with $85\,\mathrm{mm}$ aperture (i.e. the Robotiq 2F-85); these objects are transparent and were omitted in our experiments.

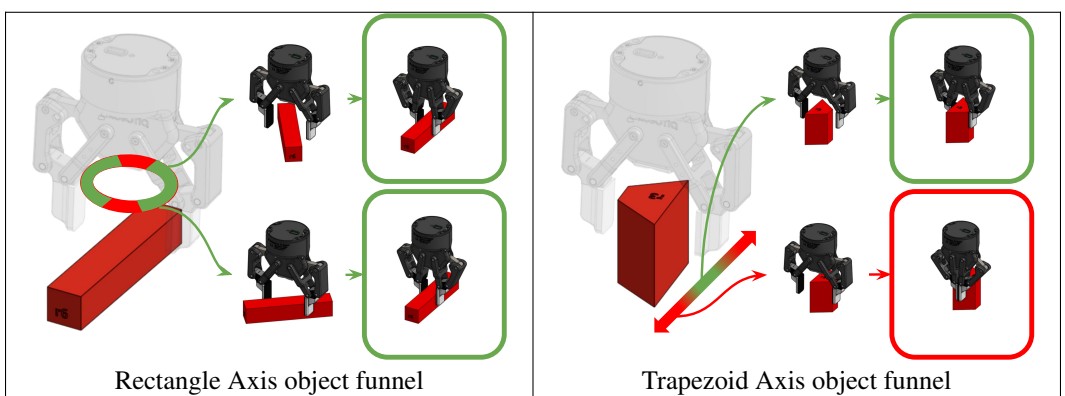

**Figure S3: A visualization of parallel-gripper grasp-affordance.** *Left:* a visualization of how the rectangle-object grasp-funnel (beam with square section) varies with the object-gripper relative orientation: successful grasps are invariant to significant orientation differences. *Right:* a visualization of how the trapezoid-object (trapezoid section) grasp-funnel varies with the object-gripper relative position: small differences in the relative pose hamper a successful grasp.

## A.2 Benchmark Analysis

The design principles outlined above aim at varying the resulting objects' grasp and stack affordance for a parallel gripper. In this section we qualitatively and quantitatively discuss these variations. The qualitative evaluations follow from some general principles on force closure[6] and object's funnel (see Mason [30] for a definition). Figure S3 shows the graphical notation which we will use to qualitatively visualize the grasp affordance with respect to the gripper-object relative pose (left) and rotation (right).

---

[6]A two-contact stable grasp is achieved if and only if Murray et al. [35, Theorem 5.6] the line connecting the contact points lies inside both friction cones (Figure S4).

**Figure S4: A sketch of the design principles adopted to vary RGB-objects' grasp affordance as a function of the applied deformation.** *Top row*: keeping in mind object funnels for the parallel gripper, different objects tolerate different 4-DoF displacements (3D Cartesian and vertical rotation) with respect to the grasping pose visualized in picture; *in green*: displacements leading to successful grasps, *in red*: displacements leading to unsuccessful grasps. *Bottom row*: a visualization of the grasp metric used in Mahler et al. [32], Wang et al. [34]. Each grasp is visualized as a line segment penetrating the object at the grasping locations; the color of the segment corresponds to the corresponding value of the Ferrari-Canny [31] epsilon grasp metric (robust grasps in green, weak grasps in red). The displayed value $\bar{Q}$ is the average of the epsilon metric for all visualized grasps.

Figure S4 shows this qualitative visualization for some representative RGB-objects: the seed object, and 4 maximally deformed objects of the 4 major axes. The visualization aims at showing that the seed cube is relatively easy to grasp and forgiving of significant errors in the gripper positioning. The Polygon axis requires a precise positioning but it's relatively robust towards errors in gripper orientation. The Trapezoid axis offers two non-parallel faces which require accuracy in both positioning and orientation. The Parallelogram axis is designed to offer the same grasp affordance of a cube but quite different stack affordance. Finally, the Rectangle axis elongates one dimension above the gripper maximum aperture thus preventing some grasps at given orientations. Interestingly, these qualitative considerations are supported by a quantitative metric, based on the Ferrari-Canny [31] epsilon grasp metric. This evaluation is shown for each of the objects considered, using the code provided in Mahler et al. [32], at the bottom row of Figure S4. A high $\bar{Q}$, used to symbolize the average metric for all grasps sampled on the object, signifies that the object evaluated is easy to grasp. A low $\bar{Q}$, as is the case e.g. with the Trapezoid, means that the object is harder for grasping.

Qualitative considerations similar to the ones done above for the grasp-affordance hold for the stack-affordance, as shown in Figure S5. In this case, the affordance is varied by having bottom objects the top flat surfaces of which differ in area, shape and orientation. The Polygon axis introduces a deformation which reduces the top surface and increases the likelihood of the object to roll; the Trapezoid axis just reduces the top surface and in some configuration doesn't afford a grasp at all requiring a re-orientation to another configuration which affords a stack; the Parallelogram axis offers a stacking surface which has an off-set with respect to the center of mass and therefore requires the top object to be carefully placed to avoid objects from tipping over; the Rectangle axis offers an augmented stacking surface along one axis and a reduced one on another axis.

We have discussed how the shapes of objects influence their grasp and stack affordances. Other interesting affordances are needed to effectively solve these tasks: (1) the clutter affordance and (2) task-oriented affordance. *Clutter* influences affordance since only a subset of grasps is feasible in presence of obstacles (right panel in Figure S6). Additionally, for some objects valid grasps are

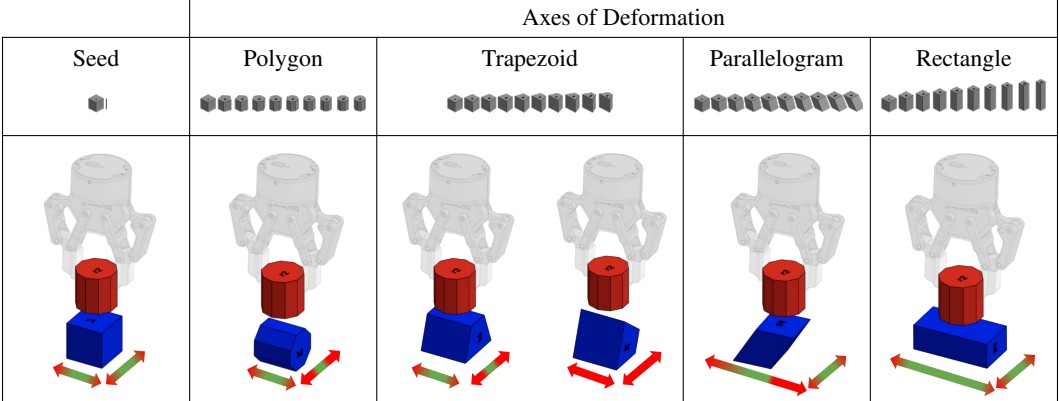

| | | Axes of Deformation | | |
| Seed | Polygon | Trapezoid | Parallelogram | Rectangle |
|---|---|---|---|---|

**Figure S5: A sketch of the design principles adopted to vary RGB-objects' stacking affordance. Bottom objects offer support of different shapes.** From left to right: curved support, small support, slanted support and wide support.

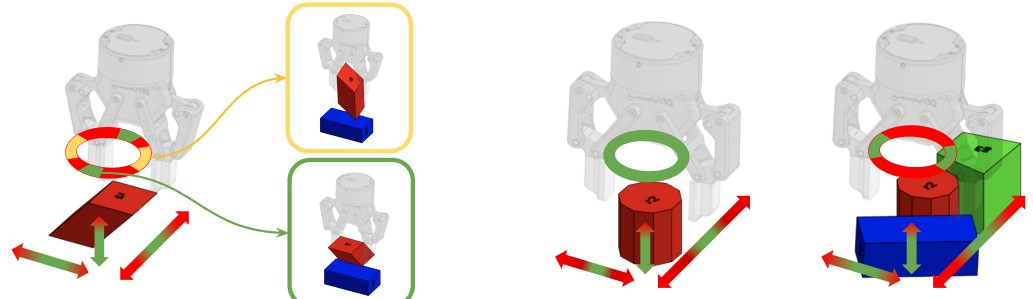

**Figure S6:** *Left:* a visualization of good stacking-oriented gripper orientations (in green) and good only-for-pick&place gripper orientations (in yellow) for the slanted cylinder. *Right*: a sketch to visualize how affordances vary in the clutter; only a subset of grasps are possible in the clutter.

not suitable for stacking and this requires our agent to perform a *task-oriented grasp* (left panel in Figure S6).

### A.3 Publicly Released Objects

Instead of standardizing the objects purchase as e.g. in Çalli et al. [36], we standardize their manufacturing procedure. We will be releasing the RGB-Objects described above and used in our experiments, depicted in their entirety in Figure S2. We decided to choose the objects to be uniform color and eventually chose only three colors: red (top objects), green (distractor objects) and blue (bottom objects). This facilitates manufacturing since each object can be manufactured with a standard 3D printer, a single filament and no additional assembly steps. Additionally, we can provide, to interested researchers, instructions on how to have such objects 3D-printed by an external vendor.

## B  Environment Details

In the following, we are going to describe the components of the robot setup that was used to conduct experiments, as well as technical considerations such as the procedure for automated evaluations, specifics of the actions and observations exposed to the agent, and reward computation.

### B.1  Real-World Environment

The environment in the real world consists of a robot arm with a gripper, a set of sensors and a basket (Figure S7), chosen both for their durability to allow continuous and autonomous operation and for safe interaction with human operators.

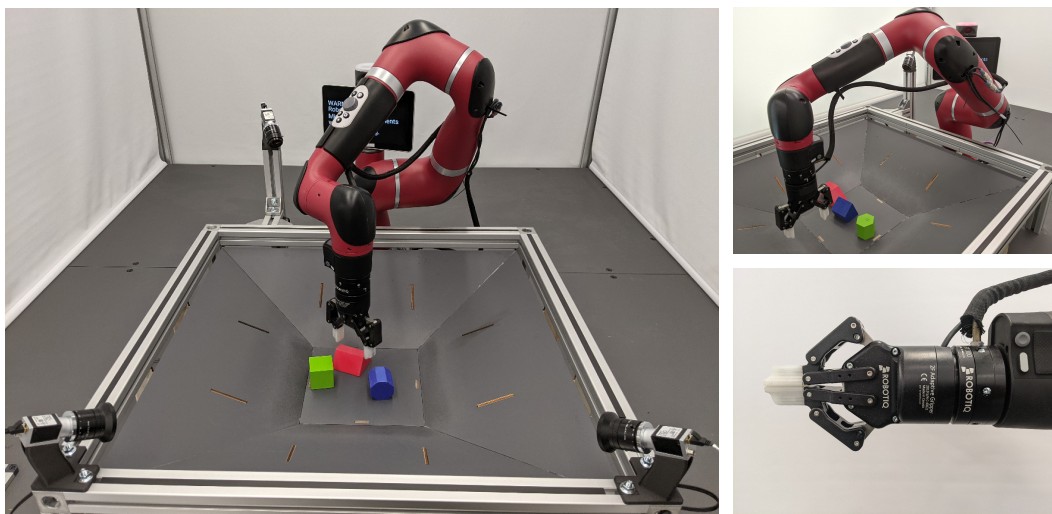

**Figure S7: Overview of the physical system.** *Left:* Front view of the basket. *Top right:* Neutral pose towards which the Cartesian controller's null-space is biased. *Bottom right:* Detail view of the endpoint tooling: force-torque sensor and gripper with custom fingertips.

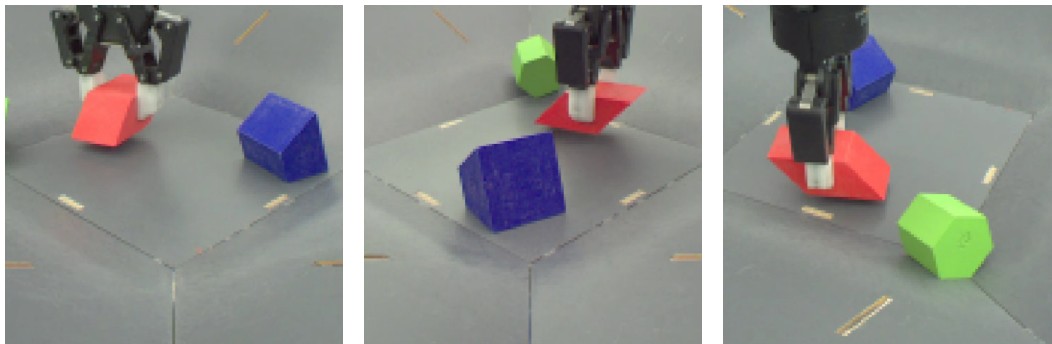

**Figure S8: Example image observations provided to the agent.** These are captured from the basket cameras, then cropped and sub-sampled to $128 \times 128$. From left to right: front left view, front right view, and back left view. In this work, we only used the front views.

*Robot arm:* assuming that exploration and manipulation require a certain level of physical interaction, we have chosen a robot capable of sensing, controlling and enduring the forces exchanged with the surroundings and the manipulated objects. We eventually selected the Sawyer from Rethink Robotics[7], both for its force and torque sensing capabilities and its use of series elastic actuators that provide passive compliance.

*Gripper:* the gripper chosen is a Robotiq 2F-85 which guarantees industrial robustness while allowing additional interaction through a passive-spring retracting degree of freedom. It is outfitted with custom fingertips printed from nylon, as the stock fingertips' rubber coating tends to wear off onto the objects and interfere with tracking.

*Sensors:* besides the torque and position sensing offered by the Sawyer, we equipped the robot with a Robotiq FT 300 force-torque sensor at the wrist. Additional perception is guaranteed by surrounding the robot with three Basler ace RGB cameras which give a complete view over the robot playground (Figure S8).

*Playground:* the basket in front of the robot, also referred to as "playground", is a laser-cut basket with slanted sides to delimit the robot's working area and to help with objects confinement. It has a $25\,\text{cm} \times 25\,\text{cm}$ bottom surface; the robot is constrained to moving its TCP inside a $20\,\text{cm}$-high virtual cube on top of this surface to ensure safe operation.

---

[7]The Sawyer is now developed and retailed by the Hahn Group.

| Component | Degrees of Freedom | Range | Unit |
|---|---|---|---|
| Cartesian translation | 3 | $[-0.07, 0.07]$ | m/s |
| Cartesian rotation (z-axis) | 1 | $[-1, 1]$ | rad/s |
| Gripper | 1 | $[-255, 255]$ | ticks[8] |

**Table S1: Ranges and units of the different components of the agent action.**

### B.1.1 Control Actions

While the robot's 7 DoFs are natively controlled in joint space, we implement a Cartesian controller to reduce the action space. We restrict the gripper to be oriented vertically, thus allowing only 4-DoF motions (3D Cartesian and 1D rotation). This restriction is used in a number of prior works studying vision-based manipulation [2, 1, 37]. A control action is fully specified by a 3D Cartesian velocity $v_x, v_y, v_z$, and an angular velocity $\omega_z$ around a vertical axis parallel to gravity. We use a P-controller to compute the horizontal angular velocity components $\omega_x$ and $\omega_y$ that keep the gripper oriented vertically, and combine it with the agent's actions to create the command for our Cartesian 6D velocity controller. Finally, a directional velocity action for the grippers single degree of freedom is added, yielding the actions summarized in Table S1.

At every environment step, after choosing an action, we then solve a constrained least-squares problem to compute the joint velocities that best realize a target Cartesian 6D velocity of the TCP [38, 39]. The null-space is controlled to bias the robot's joint positions towards a nominal configuration in the center of the joint limits. Constraints are specified to prevent the robot from violating the joint position and velocity limits, as well as avoiding collisions between the robot and the playground [40, 41]. The constrained least-squares problem is solved using an off-the-shelf QP solver [42], and the computed joint velocities are forwarded to the robot's proprietary joint velocity controller.

### B.2 Environment Observations

The robot, sensors and cameras provide various readings, which are collected at a fixed rate of 20 Hz and merged into a single observation. Not all possible observation elements are used in all stages of the system; most notably, the Cartesian object positions provided by the tracking system described in Section B.3 are only used for computing rewards, for reset, and for the scripted baselines—the learned agent cannot access this information.

We distinguish between two observation sets.

- *Full*: contains all values provided by the real robot. Parts are used for environment resets, reward computation, and scripted baselines.

- *Evaluation*: a subset of the full set, with tracker information removed.

In addition, each observation is stacked over several time steps. Observation stacking was chosen since the physical system is subject to actuation delays, and thus would not fulfil the requirements of an MDP without stacking. Camera observations are excluded from the observation stacking due to real-time and memory constraints.

The available observations, their units, and the places where each is used, are listed in Table S2. Note that all of these sets denote *available* observations, and agents can choose to omit entries; for instance, the image observation from the back camera is omitted by our agent architecture to reduce inference time.

---

[8]The gripper does not allow setting velocities in natural units, but a byte value that is mapped to a corresponding percentage of the maximum speed, which is nominally $150\,\mathrm{mm/s}$.

[9]The actual values provided by the sensor are 1 for no grasp, 2 for an inward grasp, and 3 for an outward grasp that is not possible with non-hollow objects.

[10]Measured relative to the base of the robot.

[11]This contains the previous 7-DoF joint action sent to the robot, which is distinct from the 4-DoF Cartesian action selected by the agent, and reduces ambiguity in the state.

| Observation | Unit | Size | Obs. History | Observation Set Full | Evaluation |
|---|---|---|---|---|---|
| Joint angles | rad | 7 | 3 | ✓ | ✓ |
| Joint velocities | rad/s | 7 | 3 | ✓ | ✓ |
| Joint torques | N m | 7 | 3 | ✓ | ✓ |
| Wrist pose | m, quat. | 7 | 3 | ✓ | ✓ |
| Pinch pose | m, quat. | 7 | 3 | ✓ | ✓ |
| Finger angle | ticks | 1 | 3 | ✓ | ✓ |
| Finger velocity | ticks | 1 | 3 | ✓ | ✓ |
| Grasp | discrete[9] | 1 | 3 | ✓ | ✓ |
| Wrist force | N | 3 | 3 | ✓ | ✓ |
| Wrist torque | N m | 3 | 3 | ✓ | ✓ |
| Wrist velocity | rad/s | 3 | 3 | ✓ | ✓ |
| Front left camera | RGB values | $128 \times 128 \times 3$ | 1 | ✓ | ✓ |
| Front right camera | RGB values | $128 \times 128 \times 3$ | 1 | ✓ | ✓ |
| Back left camera | RGB values | $128 \times 128 \times 3$ | 1 | ✓ | ✓ |
| Object positions[10] | m | $3 \times 3$ | 3 | ✓ | |
| Joint action[11] | rad/s | 7 | 2 | ✓ | ✓ |

**Table S2: Observations provided by the real-world robot setup.**

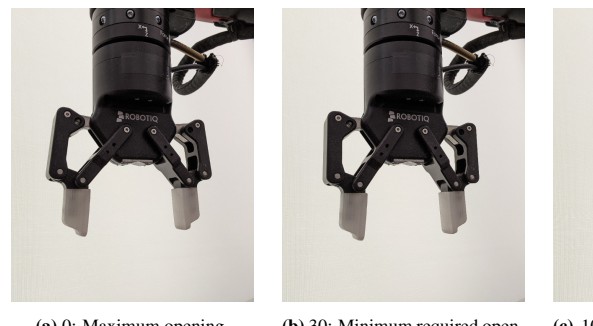
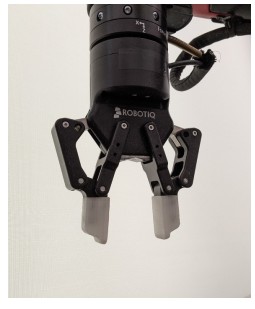
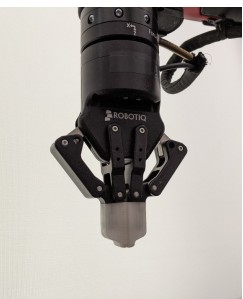

**(a)** 0: Maximum opening.  **(b)** 30: Minimum required opening to generate task reward.  **(c)** 100: Minimum opening for random initial episode states.  **(d)** 255: Fully closed.

**Figure S9: Overview of various relevant gripper positions.**

Furthermore, the Robotiq gripper used in our setup has a parallel mechanism that causes a non-linear relation between the motor encoder ticks used throughout this work, and the Cartesian distance between the fingertips. Since this relation can be hard to visualize intuitively, we present a number of poses that were used in Figure S9.

## B.3  Object Position Estimation

Several components of our setup rely on the availability of a tracking system to determine the 3D position of the objects, relative to the robot. Specifically, this is required for the scripted baseline used in this work and described in detail in Section C.2, for computing per-timestep rewards (Section B.4), and for automatically resetting the environment in automated evaluations (Section B.4.2). We therefore implemented a color-based object position estimation algorithm that provides an estimate of the 3D centroids of the red, green and blue objects. Given the critical role of this component, we calibrate (both intrinsics and extrinsics [43]) and use all three cameras available in our robot setup. The position estimation algorithm works as follows:

1. Convert the RGB into YUV images: this conversion allows finer control over colors using the chrominance components UV and robustness over brightness variations trough the luminance component Y.

2. Apply red, green and blue color masking using UV components. It is worth noting that we used the same ranges across all robot cells used in this work, while regularly applying white balancing.

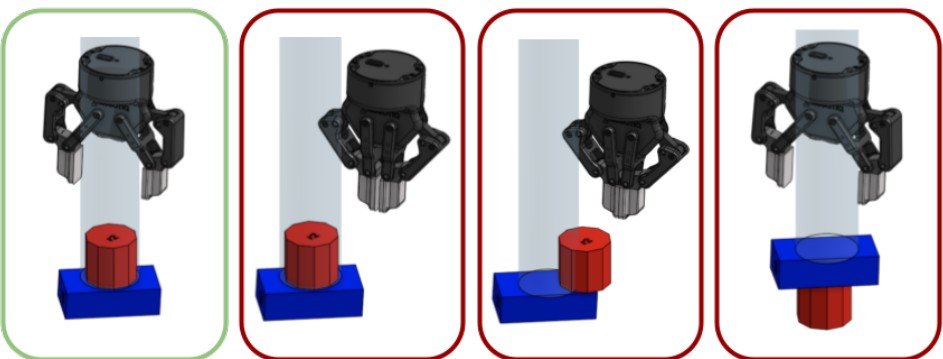

**Figure S10: Various success conditions.** From left to right: 1. Successful stack with gripper open and the top object in the cylinder region. 2. The objects are centered but the gripper is closed. 3. The top object is off-center, with its centroid outside the admissible cylinder. 4. The top object is below the cylinder region.

3. For each color, find the largest contour and evaluate its centroid using image moments [44].

4. Estimate the 3D centroids $\{[x_c, y_c, z_c]\}_{c \in \{r,g,b\}}$ of the objects in the robot reference frame through triangulation [43].

## B.4   Task Evaluation

The performance of an agent is tested on a number of trials – 200, unless specified otherwise. The initial state of the objects is randomized between episodes in the following way:

- Using a modified variant of the scripted controller (Section C.2), all objects are moved to random positions inside the working area.
- The robot's TCP is moved to a random position inside the working volume, excluding the lowest $8\,\mathrm{cm}$ to avoid collisions with objects.
- The wrist joint is rotated to a random position within $\left[-\frac{\pi}{2}, \frac{\pi}{2}\right]$.
- The fingers are moved to a random opening angle within $[0, 100]$ (i.e. open or half-open; see Figure S9 for a visualization).

Each trial lasts 20 seconds, or 400 steps at a control rate of $20\,\mathrm{Hz}$. Trials are prematurely terminated when the wrist force sensor senses horizontal forces of greater than $2\,\mathrm{N}$ or a vertical force of greater than $2.5\,\mathrm{N}$. In this case, an RL agent receives a discount of zero.

### B.4.1   Reward Definition

We all can understand what a stacked pair of objects should look like, however it is surprisingly difficult to define for a large set of geometric shapes, for the purposes of automated evaluations. Here we formalize the definition of "stacking" used in the main paper.

For objects to be considered "stacked", the top object's position (as estimated by the tracking system) must be inside a cylindrical volume of a $3\,\mathrm{cm}$ radius, starting $2.5\,\mathrm{cm}$ or higher above the bottom one, and the gripper must be fully opened. Specifically, for object centroids $[x_{top}, y_{top}, z_{top}]$ and $[x_{bottom}, y_{bottom}, z_{bottom}]$, and for finger-opening angle $f$, we define the sparse, binary, stack reward:

$$r = \begin{cases} 1, & \text{if } (z_{top} - z_{bottom} > 0.025) \wedge (||(x_{top}, y_{top}) - (x_{bottom}, y_{bottom})|| < 0.03) \wedge (f < 30) \\ 0, & \text{otherwise.} \end{cases}$$

$$(S1)$$

The open-ended cylinder was chosen to accommodate objects of arbitrary sizes. For instance, the top (red) object of Triplet 3 has a length of $15\,\mathrm{cm}$, so the centroid can be a considerable distance from the bottom object when standing on its long side. A visual depiction of different stacked pairs being considered successful (green) or failures (red) is given in Figure S10.

### B.4.2 Automation for Unattended Learning and Evaluation

The training and evaluation process used in this work is largely automated. In fact, the experiments were continued throughout multiple COVID-19 lockdowns. In particular, the randomization procedure described above is fully automatic, with a modified version of the PID controller developed for the scripted baseline (Section C.2) being used to move objects. Note that the lack of object orientation provided by the tracker means that pose randomization is not deliberate. Instead, we rely on incidental rotation when objects are dropped, and run evaluations for at least 200 trials to reduce variance.

A pool of 10 identical robot cells is used for data collection. To guarantee comparability of results, evaluations are always performed on the same five cells, each of which is associated with one specific test triplet. Furthermore, tare is performed on the wrist force-torque sensor between episodes, in order to prevent drift. All cameras are white-balanced and their brightness adjusted to the same level, to counteract both daytime fluctuations in lighting, and differences between individual robot cells.

Evaluation requests are enqueued for each cell, and processed in order of arrival; thus, there is no closed training-evaluation loop, but training from any robot data always has to be offline to some degree. In the absence of new evaluation requests, old ones are automatically repeated in order to gather additional data and reduce variance. A number of consistency checks between episodes ensure that all sensors report data at the expected rates, and that actuators are operational—failing these checks would trigger the only required human interaction.

### B.5 Simulation

Our simulation environment was implemented in the MuJoCo [45] physics simulator. Like the equivalent real robot environment, it contains a Sawyer arm with a Robotiq 2F-85 gripper mounted behind the playground, with three cameras attached to the basket.

The simulation was designed to provide the same observations as the real robot, with the same ranges and shapes. A small number of observations were too dissimilar to be of use, notably the torques, and are thus omitted. For the full list of available observations in our simulated environment at different stages of training, see Table S5. Like the observations, the simulation exposes the same 4-DoF Cartesian actions as the real robot's given in Table S1. It uses the same QP controller as described in Section B.1.1 to compute joint velocities from Cartesian velocities at 20 Hz. It was also designed to have similar appearance and dynamics to the real environment. However, as Figure S11 illustrates, the low-level appearance is noticeably different. Likewise, the physics differ in the way objects interact and slide off each other.

The evaluation protocol in simulation follows that of the real robot, with a few key differences.

1. We perform 1000 evaluation episodes per policy per Triplet[12], rather than 200.
2. The entire 6-DoF pose of the objects is randomized, rather than only the position.
3. The sparse evaluation reward makes use of privileged information from the simulator, which isn't available on the real system, specifically whether objects are directly in contact with each other. This allows us to have a wider admissible cylinder of $5\,\mathrm{cm}$ in which the top object may be placed, and eliminates the need to check the gripper's opening angle.

$$r = \begin{cases} 0, & \begin{aligned} &\text{if } ||(x_{top}, y_{top}) - (x_{bottom}, y_{bottom})|| > 0.05 \\ &\text{if } (z_{top} - z_{bottom}) < 0.02 \\ &\text{if top object is not in contact with bottom object} \\ &\text{if top object is in contact with robot or basket} \end{aligned} \\ 1, & \text{otherwise.} \end{cases} \quad (S2)$$

### B.5.1 Shaped Reward

Our shaped reward, which we designed to use for training our state-based agent *in simulation* only, forms a curriculum leading to a successful stack. It is divided into five progressive stages: **reaching**

---

[12]When evaluating on the training object set we evaluate 2 episodes for 5000 triplets.

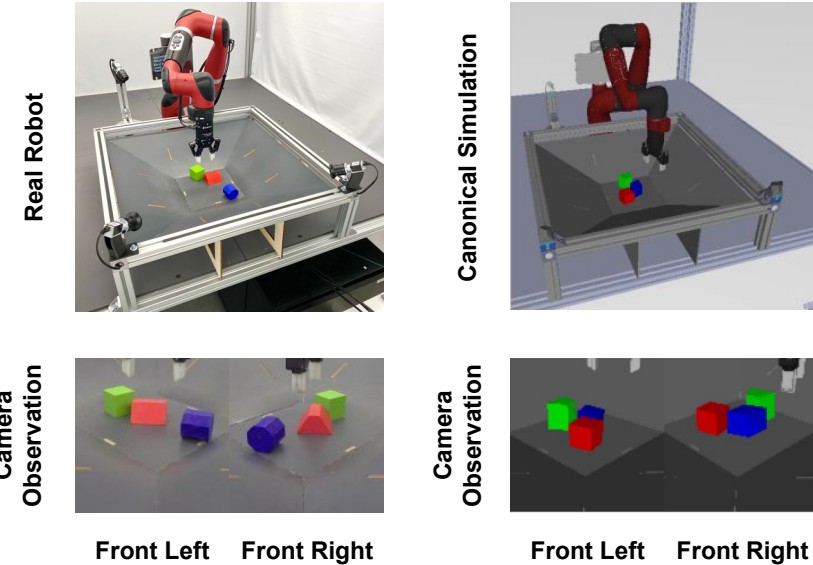

**Figure S11: Real and simulated environments.** Equivalent real and simulated environments with the camera observations used during training.

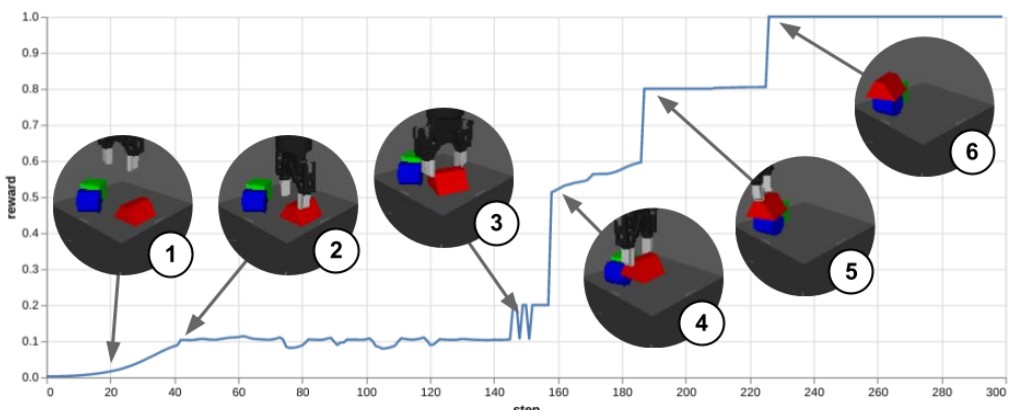

**Figure S12:** Reward trace for 300 steps of an episode. From left to right: 1. Approach during $R_{reach}$ part of $R_{grasp}$ stage. 2. $R_{close\_gripper}$ component of $R_{grasp}$ stage becomes active as the gripper is closed while realigning it to the graspable object faces. 3. Transition to $R_{lift}$ stage as the object is slightly lifted. 4. $R_{hover}$ increases as the object is moved closer to the target. 5. Objects are precisely enough placed to be considered stacked as per $R_{stack}$. 6. Gripper has moved far enough away to enter final $R_{leave}$ stage.

**and grasping** the top object, **lifting** it more than $10\,\mathrm{cm}$ above the basket, **hovering** the top object over the bottom object, **stacking** it, and **leaving** the objects stacked by moving the gripper away from them. Each stage generates a reward in $[0, 1]$, and the highest-level stage to produce a reward of $0.1$ or more is considered the "active" one.

$$r = \begin{cases} \frac{4+R_{leave}}{5}, & \text{if } R_{leave} > 0.1 \\ \frac{3+R_{stack}}{5}, & \text{if } (R_{stack} > 0.1) \wedge (R_{leave} \leq 0.1) \\ \frac{2+R_{hover}}{5}, & \text{if } (R_{hover} > 0.1) \wedge (R_{stack} \leq 0.1) \wedge (R_{leave} \leq 0.1) \\ \frac{1+R_{lift}}{5}, & \text{if } (R_{lift} > 0.1) \wedge (R_{hover} \leq 0.1) \wedge (R_{stack} \leq 0.1) \wedge (R_{leave} \leq 0.1) \\ \frac{R_{grasp}}{5}, & \text{otherwise.} \end{cases} \tag{S3}$$

Intuitively this amounts to an agent being rewarded incrementally for each of the stages that are required to complete a stable stack. A detailed description of each of the stages and the definition of the equivalent rewards can be found below. An example reward trace illustrating the different stages is also shown in Figure S12.

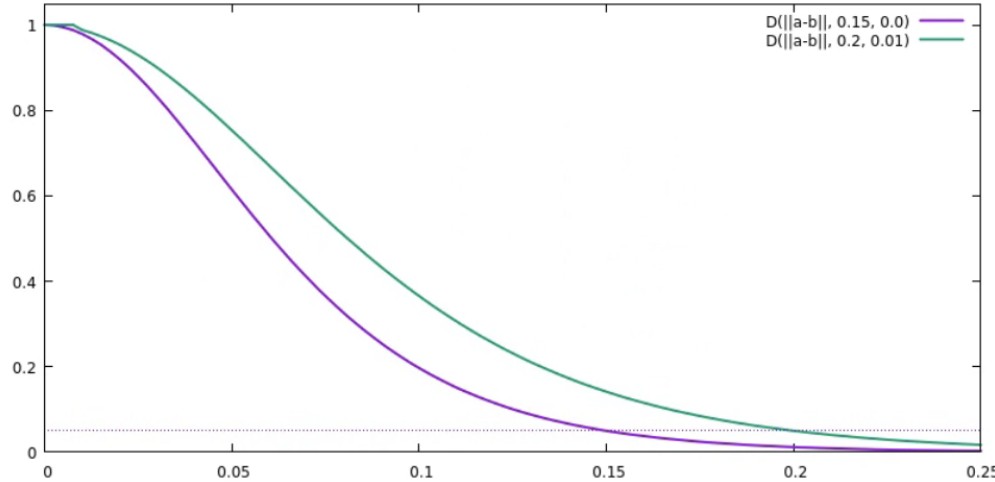

**Figure S13:** Examples of the distance function used in several reward terms, with the x-axis showing the distance between two entities $a$ and $b$. Note how the value always decays to $0.05$ (dashed line) as the distance reaches the shaping tolerance $s$.

We now describe the distinct stages of the shaped reward described above, all of which produce rewards in $[0, 1]$. As laid out in Equation (S3), the stages are combined into a total reward that consists of the "highest" of these stages that is currently generating a reward over $0.1$, plus a fixed amount for each "lower" stage.

Several of the stage rewards make use of a distance function $D(a, b, s, t)$, which is defined as the tanh over the distance between $a$ and $b$, which decays to $0.05$ as the distance reaches $s$. If the distance is below a tolerance of $t$, the maximum value of $1$ is returned. This distance function is illustrated in Figure S13.

$$D(a, b, s, t) = \begin{cases} 1, & \text{if } ||a - b|| < t \\ 1 - \tanh\left(||a - b|| \frac{\tanh^{-1}\sqrt{0.95}}{s}\right)^2, & \text{otherwise.} \end{cases} \tag{S4}$$

**Reaching and Grasping** The first stage $R_{grasp}$ provides reward when the tool center point (TCP) is moved close to the top object, with an additional bonus for closing the parallel gripper that is given only when already very close to the object.

$$R_{grasp} = R_{reach} \cdot \begin{cases} 0.5 + \frac{R_{close\_gripper}}{2}, & \text{if } R_{reach} > 0.9 \\ 0.5, & \text{otherwise.} \end{cases} \tag{S5}$$

The reaching component $R_{reach}$ is a shaped distance between the TCP position $pos_{TCP} = (x_{TCP}, y_{TCP}, z_{TCP})$ and that of the top object $pos_{top} = (x_{top}, y_{top}, z_{top})$, decaying within $15\,\text{cm}$ and with no tolerance. The positions are provided in meters with respect to the robot frame of reference, centered around the arm's base.

$$R_{reach} = D(pos_{TCP}, pos_{top}, 0.15, 0). \tag{S6}$$

The component $R_{close\_gripper}$ in turn is maximal when the grasp sensor is triggered. If not, a smaller shaped reward is given, which approaches its maximum as the gripper opening angle $f$ reaches its maximum closing angle of $255$.

$$R_{close\_gripper} = \begin{cases} 1, & \text{if grasp sensor triggered} \\ \frac{D(f, 255, 255, 0)}{2}, & \text{otherwise} \end{cases} \tag{S7}$$

**Lifting**   The lift stage $R_{lift}$ also makes use of the grasp component $R_{close\_gripper}$, but multiplies it with a shaped reward that linearly increases as the designated top object's centroid moves between a minimum height of $5.5\,\mathrm{cm}$ and a maximum of $10\,\mathrm{cm}$.

$$R_{lift} = R_{close\_gripper} \cdot R_{move\_top\_up} \tag{S8}$$

$$R_{move\_top\_up} = \begin{cases} 1, & \text{if } z_{top} > 0.1 \\ 0, & \text{if } z_{top} < 0.055 \\ \frac{z_{top} - 0.055}{0.1 - 0.055}, & \text{otherwise} \end{cases} \tag{S9}$$

where $z_{top}$ is the height of the top object from the basket.

**Hovering**   The hovering stage $R_{hover}$ simply provides reward for the top object being close to a position $4\,\mathrm{cm}$ above the bottom one. Maximum reward is given with a $1\,\mathrm{cm}$ tolerance around this position to account for noise in the tracking system. Outside this tolerance, the reward decays within $20\,\mathrm{cm}$.

$$R_{hover} = D\left(pos_{top}, pos_{bottom} + (0.0,\ 0.0,\ 0.04), 0.2, 0.01\right). \tag{S10}$$

**Stacking**   The stacking stage $R_{stack}$ is a sparse reward that is only non-zero when the red object's horizontal position is within $3\,\mathrm{cm}$ of the blue one's, and its vertical position within $1\,\mathrm{cm}$ of the point $4\,\mathrm{cm}$ above the blue one. Note that this differs from the open volume in which the red object is allowed to be (which is used for the real robot's evaluation in Equation (S1)).

$$R_{stack} = \begin{cases} 0, & \text{if } \|(x_{top}, y_{top}) - (x_{bottom}, y_{bottom})\| > 0.03 \\ & \text{if } (z_{top} - z_{bottom} + 0.04) > 0.01 \\ 1, & \text{otherwise.} \end{cases} \tag{S11}$$

**Leaving**   The final leaving stage $R_{leave}$ is identical to the stacking stage $R_{stack}$, but multiplied by a shaped term that rewards moving the TCP to a position $10\,\mathrm{cm}$ above the red object, thus forcing the agent to let go of the object. Since it is not important whether that position is precisely reached, maximum reward is given with a tolerance of $3\,\mathrm{cm}$.

$$R_{leave} = R_{stack}\ D(z_{TCP}, z_{top} + 0.1, 0.05, 0.03). \tag{S12}$$

## C   Baselines

### C.1   Human performance

As a rough indication of task difficulty, we collected a few demonstrations of the task in simulation from human teleoperators. The demonstrations were collected by 4 individuals who were not part of the research team. They used game pads to control the robot arm, and faced the same time limit as was used for evaluation of learned or scripted agents. Unlike agents, teleoperators were given a single camera view at a high resolution. Teleoperators recorded a total of 846 episodes (the number varied from 141 to 331 per participant), with the object set randomly replaced every 10 episodes.

These demonstrations were not used to train any of the agents mentioned in the paper. Results are summarized in Table S3.

### C.2   Scripted Agent

The scripted baseline is a classical robotic control approach using a lot of prior knowledge, coded in a finite-state-machine. It uses the same observations available to the agent, as well as the 3D positions of the blue and red objects' centroids. In the real environment, the 3D positions of the objects are obtained from a centroid estimation algorithm (Section B.3); in the simulated environment, the

| Objects | Success Rate | Reward | Episode Count |
|---|---|---|---|
| (Sim) Triplet 1 | 37% | 23 | 204 |
| (Sim) Triplet 2 | 36% | 16 | 160 |
| (Sim) Triplet 3 | 35% | 24 | 159 |
| (Sim) Triplet 4 | 59% | 39 | 133 |
| (Sim) Triplet 5 | 66% | 47 | 190 |

**Table S3:** Average success rate and cumulative sparse reward for each of the test object sets, from human teleoperators.

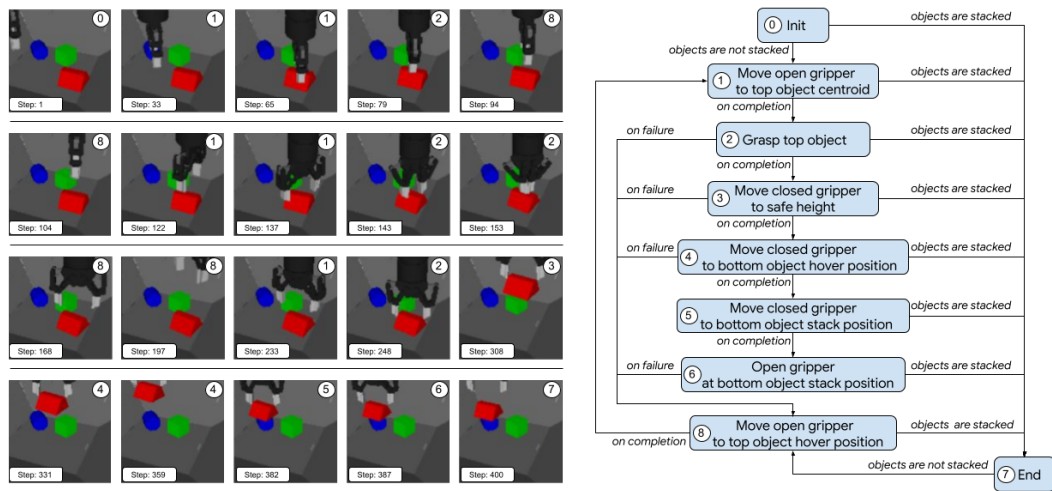

**Figure S14:** (Left) Example scripted baseline run for the test object set 1. Each figure shows the state ID on the top right, and the current step counter on the bottom left. (Right) State diagram for the finite-state machine used in the scripted baseline.

positions are obtained directly from the simulator. The performance of our scripted behaviour is meant to be used as a data point to understand how far a task-solver can get when ignoring relevant information such as the objects' orientation and shapes.

The scripted baseline was implemented through the use of a finite-state machine (FSM) with 9 states and 12 unique transition functions. A state diagram representation of the FSM is shown in Figure S14. During normal execution, the FSM starts at the 0th state and continues through states 1-6 until the objects are stacked. If the objects are found to be stacked at any point during the execution of the FSM, a transition to the final state 7 ("End") is made. If any of the states fail, the FSM immediately transitions to the 8th state, which re-positions the tool-center-point (TCP) of the robot and loops back to state 1. Note that states 0, 1, 5, and 8 do not implement transitions or failure detection and are always executed until completion.

The description of each of the states in the FSM is given below:

0. Init: No-op state that initializes the FSM and immediately transitions to the next state. This state always executes until completion;

1. Move open gripper to top object centroid: Opens the gripper and moves the TCP towards the position of the red object. Executes a non-zero angular velocity if $step > 100$, zero otherwise. Completes if the TCP is within a pre-defined threshold of the red object. This state always executes until completion;

2. Grasp top object: Closes the gripper while maintaining the TCP position close to the red object. Executes a non-zero angular velocity if $step > 100$, zero otherwise. Completes if a grasp is detected based on readings from force reading. Fails if the gripper closes and no grasp is detected;

3. Move closed gripper to safe height: Moves the TCP up while maintaining the gripper closed. Completes if the TCP is above $20\,\mathrm{cm}$. Fails if a grasp is not detected, or if the distance between the TCP and the red object becomes too large;

4. Move closed gripper to bottom object hover position: Moves the TCP to a point at an absolute height of $20\,\mathrm{cm}$ directly above the blue object while maintaining the gripper closed. Completes if the TCP is above the blue object. Fails if a grasp is not detected, or if the distance between the TCP and the red object becomes too large;

5. Move closed gripper to bottom object stack position: Moves the TCP to a point $3\,\mathrm{cm}$ above the bottom object while maintaining the gripper closed. Completes if the TCP is within a pre-defined threshold of this point. This state always executes until completion;

6. Open gripper at bottom object stack position: Opens the gripper while maintaining the TCP position $3\,\mathrm{cm}$ above the bottom object. Fails if the objects are not stacked after opening the gripper;

7. End: Opens and lifts the gripper to an absolute height of $30\,\mathrm{cm}$. Final state. Fails if the objects are not stacked;

8. Move open gripper to top object hover position: Opens the gripper and moves the TCP to a point at an absolute height of $20\,\mathrm{cm}$ directly above the red object. Executes a non-zero angular velocity if $step > 100$, zero otherwise. Completes if the TCP is above the red object at a pre-defined height. This state always executes until completion and will result in different "random" grasp orientations.

Position control of the TCP is achieved through a low-gain P-controller on the error between the desired 3D Cartesian position and the current position of the TCP. The desired position of the TCP is computed on each state individually based on the predefined behaviour of each state and the measured position of the objects through our perception pipeline. The orientation of the wrist is only actively controlled during the execution of the 1, 2, and 8th states, which execute a random angular velocity about the vertical axis after the 100th step, or zero otherwise. The outputs of the P-controller are passed directly to the first 3-DoF of the action space exposed by the environment, while the angular velocity commands (4th DoF) are set to zero during states that do not actively control the orientation.

| Objects | Success Rate | Reward |
|---|---|---|
| (Sim) Triplet 1 | 35% | 48 |
| (Sim) Triplet 2 | 30% | 58 |
| (Sim) Triplet 3 | 27% | 40 |
| (Sim) Triplet 4 | 66% | 128 |
| (Sim) Triplet 5 | 67% | 128 |
| (Real) Triplet 1 | 36% | 54 |
| (Real) Triplet 2 | 23% | 39 |
| (Real) Triplet 3 | 34% | 49 |
| (Real) Triplet 4 | 85% | 152 |
| (Real) Triplet 5 | 77% | 143 |

**Table S4:** Scripted baseline performance success rate and average cumulative reward for each of the test object sets in the simulated and real environment.

Table S4 summarizes the average performance of the scripted approach on the test sets. Each test set was evaluated for 1000 episodes in simulation, and for at least 800 episodes in the real setup. The agent achieved a success rate of 43% on the training set over $10\,000$ episodes in simulation.

## D    Methods

### D.1    Details on Training Expert policies from State Features in Simulation

As outlined in the Section 4.1, the first step in our approach is to train a policy, in simulation, either specializing on each of the 5 *fixed triplets* for the *Skill Mastery* task, or a general one on the $1\,092\,727$ triplets that are possible with the 103 training objects for the *Skill Generalization* task. As discussed,

| Observation / Reward | State-Based (Simulation) | Vision-Based (Simulation) | Vision-Based (Real) |
|---|:---:|:---:|:---:|
| Joint angles | ✓ | ✓ | ✓ |
| Joint velocities | ✓ | | ✓ |
| Joint torques | ✓ | | |
| Wrist pose | ✓ | ✓ | |
| Pinch pose | ✓ | ✓ | ✓ |
| Finger angle | ✓ | ✓ | ✓ |
| Finger velocity | ✓ | | ✓ |
| Grasp | ✓ | | ✓ |
| Wrist force | ✓ | | |
| Wrist torque | ✓ | | |
| Wrist velocity | ✓ | | |
| Front left camera | | ✓ | ✓ |
| Front right camera | | ✓ | ✓ |
| Back left camera | | | |
| Object positions | ✓ | | |
| Object pose | ✓ | | |
| Joint action | | | |
| Reward | Shaped (Simulation) Equation (S3) | Sparse (Simulation) Equation (S2) | Sparse (Real) Equation (S1) |

**Table S5: Observations and rewards used at different training stages.** The state-based teacher is trained with privileged information in simulation, and is then distilled to a vision-based policy that has access to images and only proprioception observations that are realistic in simulation and can also be later used for zero-shot sim-to-real transfer. For this reason, we exclude velocity, force, and torque observations for distillation in simulation. For the one-step offline policy improvement, a new vision-based policy is trained from a real-world dataset. This improved policy now includes velocity observations, but excludes force and torque observations since they too noisy in the real system. We did not use the back camera in our experiments.

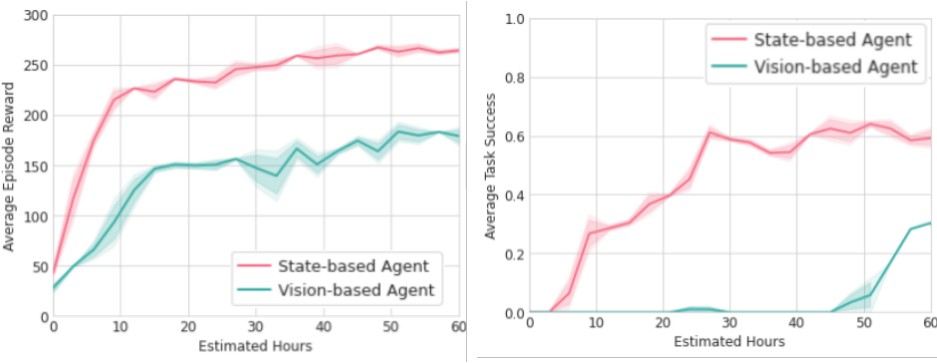

**Figure S15: State-based vs Vision-based MPO training.** Comparison of average reward (left) and average task success on the training set (right) for the Skill Generalization task when training from all available state information (State-based Agent) vs training from vision and proprioception (Vision-based Agent)

we found training directly from state features $s$ to be significantly faster than training from vision in this step and thus exposed the full simulation state—proprioceptive information from the robot and 6-DoF pose information about the objects—to the agent. The complete list of observations available to the state-based policy can be found on Table S5. At this stage of the learning pipeline, we are mainly concerned with obtaining high-performing experts in a fast manner in simulation. Thus on top of access to full state information, we also use a shaped reward which is only available in simulation and enables fast learning. The shaped reward is described in Section B.5.1 and visualized in Figure S12. We provide a comparison between training with state features vs. training directly from vision in Figure S15. As is evident from the comparison training from vision results in a large slow-down in terms of training time.

As mentioned, any off-the-shelf RL algorithm could have been used for training our state-based policies. We opted to use MPO [15], which we found to lead to fast policy improvement while allowing for stable learning. MPO does not directly optimize the RL objective, but instead considers a KL regularized objective that is optimized with a policy iteration approach. Concretely, in iteration $k$ we first learn a corresponding Q-function $Q_\phi^{\pi_{k-1}}(s, a, y)$ for the policy from the last iteration (starting from a random policy $\pi_0$ at $k_0$), which can be learned from a replay buffer $\mathcal{D}$ by finding the function that minimizes the squared temporal difference error:

$$\arg\min_\phi \mathop{\mathbb{E}}_{(s_t, a_t, s_{t+1}) \sim \mathcal{D}} \left[ \left( r(s_t) + \gamma \mathop{\mathbb{E}}_{a' \sim \pi_{k-1}(\cdot | s_{t+1}, y)} \left[ Q_{\phi'}^{\pi_{k-1}}(s_{t+1}, a', y) \right] - Q_\phi^{\pi_{k-1}}(s_t, a_t, y) \right)^2 \right],$$
(S13)

where $\phi'$ are the parameters of a *target network* [46] that are replaced with the current parameters $\phi$ for the Q-function every 200 optimization steps, and we use 20 samples from the policy to estimate the inner expectation. Instead of the single transition temporal difference error above, Abdolmaleki et al. [15] also considered a n-step temporal difference target calculated via the Retrace algorithm [47] and we use this target in our MPO implementation. This Q-function is then used to define the following KL constrained objective for policy optimization:

$$\mathcal{L}_{\mathrm{MPO}}(q) = \mathop{\mathbb{E}}_{s \sim \mathcal{D}} \left[ \mathop{\mathbb{E}}_{a \sim q} [Q^{\pi_{k-1}}(s, a, y)] \right],$$
$$\text{s.t. } \mathop{\mathbb{E}}_{\rho_{\pi_k}} [\mathrm{D_{KL}}(q(\cdot | s, y), \pi_{\theta_e}(\cdot | s, y))] < \epsilon_E,$$
(S14)

where $\mathrm{D_{KL}}$ denotes the KL divergence to the last policy, which restricts changes in the policy and induces stable learning. A solution to this problem can be found in closed form as $q(a|s, y) \propto \pi_{k-1}(a|s, y) \exp(Q^{\pi_{k-1}}(s, a, y)/\alpha)$ which can be projected back to a parametric policy by finding the expert policy $\pi_{\theta_e}$ as the maximizer

$$\pi_{\theta_e}(a|s, y) = \arg\max_{\pi_{\theta_e}} \mathop{\mathbb{E}}_{s \sim \mathcal{D}} \left[ \mathop{\mathbb{E}}_{a \sim \pi_{k-1}} [\exp(Q^{\pi_{k-1}}(s, a, y)/\alpha) \log \pi_{\theta_e}(a|s, y)] \right],$$
$$\text{s.t. } \mathop{\mathbb{E}}_{\rho_{\pi_k}} [\mathrm{D_{KL}}(\pi_{k-1}(\cdot | s, y), \pi_{\theta_e}(\cdot | s, y))] < \epsilon_M,$$
(S15)

which corresponds to minimizing the KL between $q$ and $\pi_{\theta_e}$ and where $\epsilon_M$ specifies an additional trust-region constraint placed on the policy (we set $\pi_k$ for each iteration to $\pi_{\theta_e}$ after 200 optimization steps). We use a trust-region constraint that splits the influence on the mean and covariance for Gaussian policies as in Abdolmaleki et al. [48]. When using the hybrid space of continuous and discrete actions, we have a separate third trust-region constraint for the Bernoulli distribution, though we found that the discrete component didn't require a trust-region for stable learning. Optimization can be carried out via Monte-Carlo estimation of the objective using samples from the policy $\pi_k$ to estimate the inner expectation, and samples from the replay buffer for the outer expectation. For a full description of the algorithmic details of solving this optimization problem we refer to Abdolmaleki et al. [15, 48].

### D.2 Details on Interactive Imitation Learning for Sim-to-Real Transfer

After obtaining the experts via MPO, we distill the state-based experts into a single vision-based policy $\pi_\theta^{\mathrm{vis}}$ via interactive imitation learning, as described in Section 4.2. In this step the $\pi_\theta^{\mathrm{vis}}$ uses only a subset of the available observations (vision and proprioceptive readings but no information about object positions; see Table S5 for a complete list). The two key decisions made for distillation are: 1) We collect data using $\pi_\theta^{\mathrm{vis}}$ while it is being trained. For this purpose we run a large number of

"actor" processes (1000) in simulation with the domain randomized environment. These fetch the parameters $\theta$ from the learner process at the beginning of every episode and send data to a replay buffer $\mathcal{D}_{\text{s2r}}$. 2) We train $\pi_\theta^{\text{vis}}$ based on feedback from the expert's on data sampled from the replay (this DAgger style training resulted in best performance as outlined in the experiments). We note that this is a purely supervised learning problem on a changing dataset; as is standard the influence of $\pi_\theta^{\text{vis}}$ on the dataset collection process is only implicit (i.e. we do not calculate the gradient of the sampling process for data-collection).

### D.3 Details on Training Improved Policies from Real Data

When training improved policies from *real* data collected by executing $\pi_\theta^{\text{vis}}$ on the real robots, we use a slightly different subset of observations for the improved vision-based policy $\pi_\theta^{\text{imp}}$ (now including velocity information; see Table S5 for a complete list). As described in Section 4.3, we use a filtered cloning loss of the data for this purpose, with filtering function $f(s_t, a_t, \tau)$ where $\tau$ corresponds to the trajectory data from the executed episode. When using BC-IMP, we simply set $f(s_t, a_t, \tau) = r(s_T)$, i.e. it is 1 if the binary sparse reward of the last step in the episode (at time $T$) is 1, and 0 otherwise. This sparse reward information is readily available from the recorded episodes. For the exponential advantage filter (i.e. CRR-IMP), we use $f(s_t, a_t, \tau) = \exp(A^{\pi_\theta^{\text{imp}}}(o(s_t), a_t)/\alpha)$, in which case we need to learn an estimate of the advantage alongside the policy $\pi_\theta^{\text{imp}}$. We follow the implementation of CRR [25] and learn a distributional action-value function [49] from the same data that the policy is learned from by gradient descent on the objective:

$$\mathcal{L}_{\text{CRR-IMP}}^Q(\phi) = \mathop{\mathbb{E}}_{(s_t,a_t,s_{t+1})\sim\mathcal{D}_{\text{real}}}\left[D\left(r(s_t) + \gamma \mathop{\mathbb{E}}_{a'\sim\pi_\theta^{\text{imp}}(\cdot|o(s_{t+1}))}[Q_{\phi'}(o(s_{t+1}), a')], Q_\phi(o(s_t), a_t)\right)\right],$$
(S16)

where $D$ denotes the distributional Q-learning operator, $\phi'$ denotes the parameters of a target network (that are swapped for $\phi$ every 200 optimization steps) and where $Q_\phi(o(s_t), a_t)$ now is parameterized as a categorical distribution with 101 categories representing equally spaced bins of values from $[-150.0, 150.0]$. We learn this Q-function alongside the policy, and use $Q_{\phi'}$ to calculate policy improvement (i.e. the advantage used in the exponential filter is also fixed for 200 optimization steps at a time). We calculate the advantage $A^{\pi_\theta^{\text{imp}}}(o(s_t), a_t)$ using a Monte-Carlo estimate of

$$A^{\pi_\theta^{\text{imp}}}(o(s_t), a_t) = Q_{\phi'}(o(s_t), a_t) - \mathop{\mathbb{E}}_{a'\sim\pi_\theta^{\text{imp}}(\cdot|o(s_t))}[Q_{\phi'}(o(s_t), a')]$$

where we estimate the expectation with 20 samples from $\pi_\theta^{\text{imp}}(\cdot|o(s_t))$.

## E  Experimental Details

### E.1  Domain Randomization and Image Augmentation

As mentioned above, our strategy for solving the RGB-stacking tasks in the real world is simulation-to-reality transfer. It is therefore of paramount importance to ensure that both stages described above will result in policies that are able to bridge the reality gap and perform well in our real-world setup. We do so by relying on *(a)* a simulation environment that is closely aligned to the real robot environment (in terms of camera poses, robot joint limits, etc.); and *(b)* a sufficient amount of domain randomization [19, 24] and visual data augmentation [50]. These ensure that the simulation-trained policies can successfully deal with the domain gap that still exists between the simulated and the real environments, and the increased stochasticity of the real world. Although we did consider learned adaptation methods as used in prior work [4, 51, 52] like using domain-adversarial losses [53, 54] and randomized-to-canonical networks [55], preliminary results did not seem to provide clear benefits on top of domain randomization and data augmentation.

### E.1.1  Domain Randomization

Domain Randomization (DR) has been shown to be a simple and powerful method to achieve generalization of simulation-trained policies to the real world for robotic learning problems [56, 19, 4, 13]. In our simulated environment we randomized a number of physical properties (e.g. mass, friction, damping, armature) for all agents, as well as the delay of executing their actions on the environment.

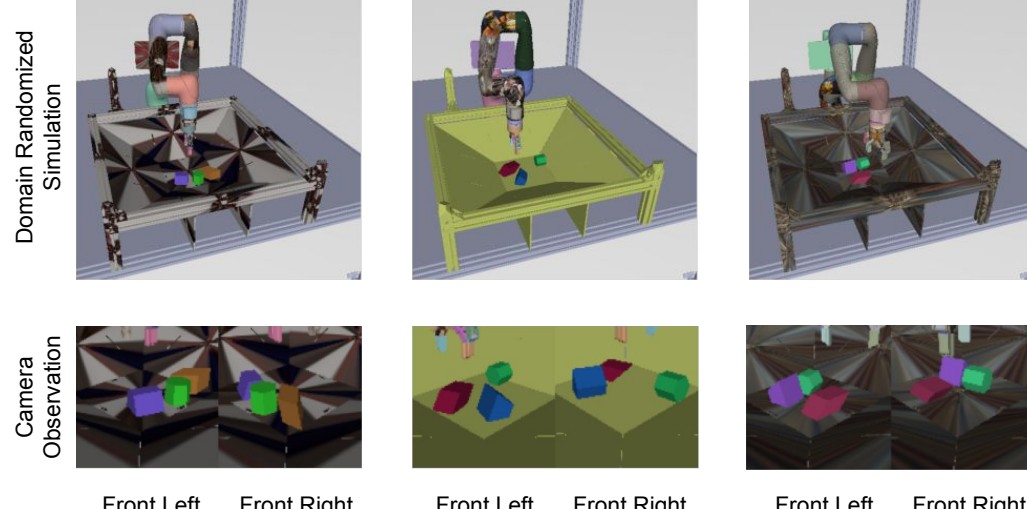

Front Left     Front Right       Front Left     Front Right       Front Left     Front Right

**Figure S16:** Visual illustration of our visual domain randomization with different sampling of the properties listed in Table S6 for the same set of objects - Triplet 2. Although all geometries can vary freely, the RGB-objects are restricted to a certain range "around" red, blue, and green to aid with the identification of these colors in the real world as well, as that is the way the vision-based agent knows which objects is the top, the bottom and the distractor.

We also randomized a number of visual properties (e.g. object colors, object textures, camera poses, lighting) for our vision-based agents at the distillation phase. In our randomized environments we uniformly sample, from pre-defined ranges, colors and textures for all geometries in our simulator, as well as lighting, and camera poses to create a large visual diversity. Action execution was randomly delayed $0, 1$ or $2$ timesteps, the equivalent of $0, 50$ or $100\,\mathrm{ms}$. Most physics properties did not require any particular tuning and are perturbed uniformly within $\pm 10\%$ of their default values in the non-randomized version of our environment. The only exceptions are the ranges for the tangential, torsional, and rolling friction of the gripper, which were tuned carefully to prevent unrealistic grasping behaviours, e.g. grasping and lifting an object by a corner. The ranges for these were determined by teleoperating the simulated robot with different friction values. A list of all properties randomized, along with the range these were uniformly sampled from, can be found on Table S6. A few samples illustrating our object color and texture randomization can be seen in Figure S16. Note that MuJoCo multiplies the RGBA values if both *texture* and *rgba* properties were set, which results in undesirably dark appearance. Thus, for each geom, we alternate sampling textures or colors. Our RGB-objects were treated differently in order to maintain their basic color, as the task is defined based on the color theme of the objects. Firstly, they are never assigned a texture. Secondly, the hue range of each object is predefined in a way that maintains the color theme, and for each RGB-object we sample the color in HSV space.

### E.1.2   Image Augmentation

In order to further increase the diversity of the data and the zero-shot real-world performance of our simulation-only trained agents, we applied a number of image transformations to our visual observations on top of domain randomization. In addition, we applied the same image transformations when directly training from real-world data (e.g. for policy improvement). Unlike domain randomization, image augmentation is applied directly on image observations, so it is applicable to images from both simulated data and real-world data.

The following transformations are applied for image augmentation: random brightness, random hue, random saturation, random contrast, and random translation. These transformations are applied sequentially in that order. The random translations use bilinear interpolation and "reflect" fill mode (i.e. the input is extended by reflecting about the edge of the last pixel). For temporal consistency, we sample the random augmentations and apply the same random offsets for all images within a trajectory subsequence. A list of the image augmentation properties, along with the ranges these

| Property | Range |
|---|---|
| RGB | $[(128, 128, 128), (255, 255, 255)]$ RGB |
| Texture | set of 117 textures |
| Red object color | $[(-35, 0.5, 0.5), (35, 1, 1)]$ HSV |
| Green object color | $[(95, 0.5, 0.5), (165, 1, 1)]$ HSV |
| Blue object color | $[(200, 0.5, 0.5), (270, 1, 1)]$ HSV |
| Ambient light color | $(0.3, 0.3, 0.3) \cdot (1 \pm 0.1)$ RGB |
| Diffuse light color | $(0.6, 0.6, 0.6) \cdot (1 \pm 0.1)$ RGB |
| Camera front left position | $(1, -0.395, 0.253) \cdot (1 \pm 0.1)$ m |
| Camera front left Euler | $(1.142, 0.004, 0.783) \cdot (1 \pm 0.05)$ rad |
| Camera front right position | $(0.967, 0.381, 0.261) \cdot (1 \pm 0.1)$ m |
| Camera front right Euler | $(1.088, 0.001, 2.362) \cdot (1 \pm 0.05)$ rad |
| Camera field of view | $[30, 40]$ |
| Gripper friction coefficient | $[(0.3, 0.1, 0.05), (0.6, 0.1, 0.005)]$ |
| Hand friction coefficient | $(1.0, 0.005, 0.0001) \cdot (1 \pm 0.1)$ |
| Arm friction coefficient | $(0.1, 0.1, 0.0001) \cdot (1 \pm 0.1)$ |
| Basket friction coefficient | $(1.0, 0.001, 0.001) \cdot (1 \pm 0.1)$ |
| Objects friction coefficient | $(1.0, 0.005, 0.0001) \cdot (1 \pm 0.1)$ |
| Objects mass | $0.201 \cdot (1 \pm 0.1)$ kg |
| Arm joint armature | $1.0 \cdot (1 \pm 0.1)$ kg m$^2$ |
| Hand driver joint armature | $0.1 \cdot (1 \pm 0.1)$ kg m$^2$ |
| Arm joint damping | $0.1 \cdot (1 \pm 0.1)$ N s/m |
| Hand driver joint damping | $0.2 \cdot (1 \pm 0.1)$ N s/m |
| Hand spring link joint damping | $0.00125 \cdot (1 \pm 0.1)$ N s/m |
| Arm joint friction loss | $0.3 \cdot (1 \pm 0.1)$ kg/(m$^2$ s$^2$) |
| Actuator gear | $(1, 0, 0, 0, 0, 0) \cdot (1 \pm 0.1)$ |
| Action delay | $[0, 2]$ timesteps |

**Table S6: Domain randomization properties that are randomized in simulation and their ranges.** These properties were sampled uniformly at the beginning of every episode.

| Property | Range |
|---|---|
| Brightness | $[-{}^{32}/_{255}, {}^{32}/_{255}]$ |
| Hue | $[-{}^{1}/_{24}, {}^{1}/_{24}]$ |
| Saturation | $[0.5, 1.5]$ |
| Contrast | $[0.5, 1.5]$ |
| Translation (horizontal and vertical) | $[-4, 4]$ pixels |

**Table S7: Image augmentation properties that are randomized and their ranges.** We sample random offsets from these ranges and apply the same random offsets to the entire sampled trajectory subsequence. We resample the offsets for each subsequence in the batch. That is, the random augmentations are consistent across time, but not across the batch.

were uniformly sampled from, can be found on Table S7. We chose these ranges qualitatively without any tuning for evaluation success. For the random perturbations of the hue, we chose ranges small enough so that the red, green, and blue objects stay reasonably close to their respective colors. A few samples illustrating the effect of image augmentation can be seen in Figure S18 for images from the real robots, Figure S19 for canonical images from simulation, and Figure S20 for domain-randomized images from simulation.

## E.2 Additional Network Architecture Details

The inputs to the networks are preprocessed in the same way for all the networks. The image observations are normalized to $[0, 1]$, whereas the non-image observations are flattened and concatenated into a single vector. The actions are normalized to $[-1, 1]$. The networks operate with normalized actions, i.e. critic networks processes normalized actions as inputs, and actor networks output action distributions in the normalized space. The agent scales back the actions to the original space when executing them in the environment.

| Hyperparameter | Value |
|---|---|
| actor network | |
|     input normalizer size | 512 |
|     MLP sizes | (512, 512, 256, 256) |
|     MVN distribution $\sigma_{\min}$ | $10^{-4}$ |
|     activations | ELU |
| critic network | |
|     input normalizer size | 512 |
|     MLP sizes | (512, 512, 256) |
|     activations | ELU |

**Table S8: Network Architecture Hyperparameters for State-Based Agents.**

In both the state-based and vision-based agents, we use an input normalization layer for the non-image observations. This input normalization layer consists of a linear layer, layer norm layer, and a tanh non-linearity. The size of the input normalizer refers to the number of units of the linear layer.

We use an output distribution layer for the output of the actor networks. The output distribution layer for the actor network outputs an independent joint distribution of multivariate normal (MVN) distribution with diagonal variance for the continuous action dimensions, and a Bernoulli distribution for the binary action dimension. The mean of the MVN is the output of a linear layer and the diagonal standard deviation is the output of a fully-connected layer with softplus non-linearity plus a bias of $\sigma_{\min}$. This distribution is not constrained to output normalized actions in $[-1, 1]$; instead, we clip samples from this distribution depending on the context. The logits vector of the Bernoulli distribution is the output of a linear layer with output size 2. This distribution is scaled accordingly to output normalized actions in $\{-1, 1\}$.

**State-based agents.** The actor network consists of an input normalization layer, MLP, and output distribution layer. The critic network starts with an input normalization layer for the observations and clipping of the actions to $[-1, 1]$, then both streams are concatenated, and followed by an MLP and a linear layer with 1 output. These MLPs use exponential linear unit (ELU) activations. See Table S8 for a full list of network architecture hyperparameters used for the state-based agents.

**Vision-based agents.** The actor network consists of an observation encoder, MLP, Transformer, another MLP, and output distribution layer. When using a critic (i.e. in CRR), the critic network starts with an observation encoder for the observations and clipping of the actions to $[-1, 1]$, then both streams are concatenated, and followed by an MLP and discrete-valued output distribution layer.

The actor and critic networks use the same architecture for their observation encoders, but their parameters are not shared. The observation encoder consists of two parallel streams—a ResNet stack for the image observations and an MLP with a final activation for the proprioception observations—and the outputs are merged by concatenation. The ResNet stack consists of a pair of ResNet encoders (one for each of the two images), activation, flattening and concatenation (of encodings from both images), and MLP with a final activation. Each ResNet encoder consists of 3 ResNet group modules. Each group module first applies a convolution followed by downsampling with a max-pooling layer, and then applies residual blocks modules twice. Each residual block consists of 2 convolution layers interleaved with non-linear activations.

The output distribution layer for the critic network outputs a discrete-valued distribution with support $[v_{\min}, v_{\max}]$ that is uniformly spaced among $n_{\text{atoms}}$ atoms or bins. The logits vector of this distribution is the output of a linear layer with output size $n_{\text{atoms}}$.

See Table S9 for a full list of network architecture hyperparameters used for the vision-based agents. We found in preliminary experiments that having each image processed by a ResNet encoder led to better sim-to-real transfer performance compared to stacking the two images along the channel dimension and processing this stacked image with a single ResNet encoder. We also found that sharing the parameters for the ResNets of both images led to even better transfer performance. This parameter sharing also has an additional advantage of computational speedups (this can be achieved by applying the ResNet in a single pass to both images concatenated along the batch dimension).

| Hyperparameter | Value |
|---|---|
| image observation encoder (actor and critic) | |
| individual ResNet per image? | yes |
| share parameters for the ResNets of both images? | yes |
| ResNet number of channels for each group | (64, 128, 256) |
| ResNet number of blocks per group | 2 |
| ResNet convolution kernel size | $3 \times 3$ |
| ResNet max-pooling size | $3 \times 3$ |
| ResNet max-pooling stride | 2 |
| post-ResNet MLP sizes | (256) |
| activations | ReLU |
| proprioception observation encoder (actor and critic) | |
| input normalizer size | 256 |
| MLP sizes | (256) |
| activations | ReLU |
| actor network (after the observation encoder) | |
| pre-Transformer MLP sizes | (512, 512) |
| Transformer number of heads | 4 |
| Transformer number of layers | 1 |
| Transformer value size | 64 |
| Transformer memory size | 8 |
| post-Transformer MLP sizes | (256, 256) |
| —MLP sizes for ablation without Transformer | (1024, 512, 512, 256, 256) |
| MVN distribution $\sigma_{\min}$ | $10^{-4}$ |
| activations | ELU |
| critic network (after the observation encoder) | |
| MLP sizes | (512, 512, 256) |
| discrete-valued distribution number of atoms $n_{\text{atoms}}$ | 101 |
| discrete-valued distribution range of values $[v_{\min}, v_{\max}]$ | $[-150.0, 150.0]$ |
| activations | ELU |

**Table S9: Network Architecture Hyperparameters for Vision-Based Agents.** The critic hyperparameters are only applicable to the methods that use a critic, i.e. CRR. Although the actor and critic networks use the same observation encoder *architecture*, their parameters are not shared.

Note that although the observation encoders use rectified linear unit (ReLU) activations throughout, the subsequent MLPs use ELU activations.

**Transformer Architecture.** While image augmentation and domain randomization helps with bridging visual and physics domain gap, the gap of transition dynamics remains. A potential approach is to give agents access to temporal information, which encourages them to "reason" about the transition dynamics. It has been shown in prior work on simulation-to-reality transfer [13], that doing so can bridge the reality gap further, and might even enable the agents to be performing system identification that can help with transfer in previously unseen environments. The Transformer architecture has been widely adopted for natural language processing [57, 33] as well as for computer vision [58]. However its application to control problems remains limited. Thereby we explore the Transformer model, which demonstrated huge power of sequence based data processing with attention mechanism, to encode temporal information. In this work, we adapted to the Transformer-XL [33]. The transformer network has stacked self-attention module that apply to the input sequence repeatedly. The transformer module consists of 1) a multi-head attention submodule followed by 2) a multi-layer perceptron network.

The transformer torso takes encoded observations as input. The multi-head attention module applies scaled dot-production attention for every timestep:

$$\text{Attention}(Q, K, V) = \text{softmax}\left(\frac{QK^T}{\sqrt{d_k}}\right) V \tag{S17}$$

| Hyperparameter | Value |
|---|---|
| **MPO** | |
| discount factor $\gamma$ | 0.99 |
| actions sampled per state | 20 |
| KL constraint $\epsilon_E$ on non-parametric policy | 0.1 |
| trust-region $\epsilon_M$ on policy (MVN mean) | $5 \times 10^{-3}$ |
| trust-region $\epsilon_M$ on policy (MVN covariance) | $1 \times 10^{-4}$ |
| trust-region $\epsilon_M$ on policy (Bernoulli) | none |
| update period for target networks | 200 |
| **General** | |
| batch size | 512 |
| trajectory length | 10 |
| environment frames per gradient step | 250 |
| replay buffer size | $2 \times 10^6$ |
| optimizer | Adam [59] |
| actor initial learning rate | $5 \times 10^{-5}$ |
| actor learning rate decay factor | 0.9 |
| actor learning rate decay schedule | range(1000000, 5000000, 300000) |
| critic initial learning rate | $1 \times 10^{-4}$ |
| critic learning rate decay factor | 0.9 |
| critic learning rate decay schedule | range(1000000, 5000000, 300000) |
| dual learning rate | $1 \times 10^{-2}$ |

**Table S10: Hyperparameters for Training State-Based Agents.**

In addition to perform a single attention operation, it is beneficial to project $Q, K, V$ $h$ times with learned linear projections respectively, where $h$ is number of heads:

$$\text{MultiHeadAttention}(Q, K, V) = \text{Concat}(\text{head}_1, ..., \text{head}_h)W^o \tag{S18}$$

$$\text{head}_i = \text{Attention}(QW_i^Q, KW_i^K, VW_i^V) \tag{S19}$$

Following the multi-head attention module, a residual connection and layer normalization are applied. To leverage the order of the sequence, a positional-encoding layer is added to the input embeddings. A fixed positional encoding using sine and cosine functions of various frequencies are used in this work:

$$\text{SelfAttention}(Q, K, V) = \text{LayerNorm}(\text{MultiHeadAttention}(Q, K, V) + \text{PositionalEncoding}(V)) \tag{S20}$$

On top of the self-attention layer, a fully-connected feed-forward network is applied to the output of each timestep separately. The MLP consists of two linear layers with a ReLU activation in between.

## E.3 Additional Training Details

We use a trust-region constraint for the policy update in MPO and CRR. Similarly to Abdolmaleki et al. [15], the mean and standard deviation of the multivariate normal (MVN) of the actor distribution have separate trust-region constraints. However, we do not use a trust-region constraint for the Bernoulli distribution component as we found it was not necessary for stable learning.

We use different learning rates for optimizing the actor, critic, and dual variables. We anneal the learning rates for the actor and the critic, following an exponential decay schedule denoted as $\text{range}(i_{\text{start}}, i_{\text{end}}, i_{\text{step}})$, where $i_{\text{start}}$ and $i_{\text{end}}$ indicate the gradient step iteration at which the annealing starts and ends, respectively, and $i_{\text{step}}$ indicates the interval at which the learning rate is multiplied by the decay factor.

See Table S10 and Table S11 for a full list of training hyperparameters used for state-based and vision-based agents, respectively.

## E.4 Detailed Real-Robot Results

In Table 2 and Table 3 we provide, for each setting, averages of multiple runs. These are 4 runs for sim-trained distilled agents: 2 seeds for each of the 2 state-based teacher policies we distilled these

| Hyperparameter | Value |
|---|---|
| **IIL (DAgger) and BC** | |
| loss | negative log-likelihood |
| —loss for ablation using MSE loss | mean squared error |
| **CRR** | |
| discount factor $\gamma$ | 0.99 |
| actions sampled per state | 20 |
| KL constraint $\epsilon_E$ on non-parametric policy | 0.1 |
| trust-region $\epsilon_M$ on policy (MVN mean) | 0.1 |
| trust-region $\epsilon_M$ on policy (MVN covariance) | 0.1 |
| trust-region $\epsilon_M$ on policy (Bernoulli) | none |
| update period for target networks | 200 |
| **General** | |
| batch size | 64 |
| trajectory length | 10 |
| environment frames per gradient step (online) | 250 |
| replay buffer size (online) | $1 \times 10^5$ |
| dataset size (offline) | 85 213 episodes, 58 979 successful (Skill Mastery) 38 446 episodes, 14 381 successful (Skill Generalization) |
| optimizer | Adam [59] |
| actor initial learning rate | $1 \times 10^{-4}$ |
| actor learning rate decay factor | 0.9 |
| actor learning rate decay schedule | range(25000, 1000000, 25000) (Skill Mastery) |
| critic initial learning rate | $2 \times 10^{-4}$ |
| critic learning rate decay factor | 0.9 |
| critic learning rate decay schedule | range(25000, 1000000, 25000) |
| dual learning rate | $1 \times 10^{-2}$ |

**Table S11: Hyperparameters for Training Vision-Based Agents.** The critic hyperparameters are only applicable to the methods that use a critic, i.e. CRR. We found that CRR doesn't need a tight trust-region for stable learning, so we chose a loose constraint of 0.1 without further tuning. The number of environment frames per gradient step and replay buffer size are only applicable in the online setting, which uses the simulated environment. The dataset size is only applicable in the offline setting, and the dataset consists of real-world episodes collected on the robots. The dataset for Skill Mastery only has the test triplets and the dataset for Skill Generalization only has the training objects.

from, and 2 runs for vision agents trained from real data. Here, we provide the results for each of the runs in each setting, and also present the results for each specific triplet. We hope that this provides a better sense of the variance in each setting. Entries in Table S12 correspond to Table 2, whereas entries in Table S13 correspond to Table 3.

### E.5   Qualitative Analysis

In the main text we described different challenges posed by the test triplets. Aside from the quantitative success scores above, it is therefore also interesting to look at the resulting policies qualitatively, and examine whether our agent visibly learns to overcome these challenges. Even if such an approach is naturally to be taken only as anecdotal, it nevertheless provides an indication of open challenges that are worth pursuing further.

We therefore performed a number of evaluations with the best-performing agent, the Skill Mastery CRR-IMP policy trained on sim-to-real agent data (see Table S13). We adversarially moved the objects to generate challenging situations and observed the agent's behaviour.

**Triplet 1.** The main challenge of this triplet is the need to precisely orient the gripper, since closing them on the slanted sides of the object will fail. The agent exhibits this behaviour, waiting to close the gripper until the wrist is properly aligned (Figure S17(a)).

**Triplet 2.** The bottom object in this triplet can be oriented in such a way that its top surface is slanted, making it impossible to stack without first tipping it over. The agent can be seen to perform this kind of behaviour, although not perfectly reliably; if the object is already oriented so that it can be tilted by pushing it against the basket's slope, it will do so (Figure S17(b)). However, if the

| Method | Run | Simulation Success | | | | | | | Real-Robot Success | | | | | |
|---|---|---|---|---|---|---|---|---|---|---|---|---|---|---|
| | | Training Objects | Triplet Avg. | Triplet 1 | Triplet 2 | Triplet 3 | Triplet 4 | Triplet 5 | Triplet Avg. | Triplet 1 | Triplet 2 | Triplet 3 | Triplet 4 | Triplet 5 |
| *Skill Mastery* | | | | | | | | | | | | | | |
| IIL-s2r (deterministic) | Teacher 1 - Seed 1 | N/A | **72.8%** | 74.7% | 47.5% | 82.5% | 74.4% | 84.8% | **68.1%** | 74.5% | 49.0% | 59.5% | 87.0% | 70.5% |
| IIL-s2r (deterministic) | Teacher 2 - Seed 1 | N/A | **71.2%** | 74.4% | 45.6% | 81.6% | 69.0% | 85.3% | **73.7%** | 78.0% | 64.0% | 69.5% | 86.0% | 71.0% |
| IIL-s2r (deterministic) | Teacher 1 - Seed 2 | N/A | **71.4%** | 75.2% | 45.3% | 79.9% | 72.6% | 84.2% | **66.1%** | 67.5% | 49.0% | 57.5% | 82.0% | 74.5% |
| IIL-s2r (deterministic) | Teacher 2 - Seed 2 | N/A | **71.5%** | 76.0% | 44.1% | 80.9% | 71.8% | 84.7% | **63.7%** | 71.0% | 33.0% | 57.5% | 85.5% | 71.5% |
| IIL-s2r (stochastic) | Teacher 1 - Seed 1 | N/A | **75.1%** | 80.6% | 50.7% | 82.1% | 74.9% | 87.2% | **66.7%** | 71.0% | 47.0% | 58.5% | 86.0% | 71.0% |
| IIL-s2r (stochastic) | Teacher 2 - Seed 1 | N/A | **73.3%** | 72.3% | 49.1% | 82.6% | 75.7% | 86.7% | **63.6%** | 60.5% | 40.0% | 62.5% | 80.5% | 74.5% |
| IIL-s2r (stochastic) | Teacher 1 - Seed 2 | N/A | **74.7%** | 77.1% | 54.4% | 81.2% | 74.5% | 86.6% | **70.1%** | 81.0% | 48.5% | 63.0% | 83.5% | 74.5% |
| IIL-s2r (stochastic) | Teacher 2 - Seed 2 | N/A | **73.7%** | 74.2% | 51.0% | 82.6% | 74.4% | 86.2% | **69.1%** | 80.5% | 54.5% | 54.0% | 88.0% | 68.5% |
| No Transformer | Teacher 1 - Seed 1 | N/A | **74.3%** | 74.9% | 52.9% | 82.6% | 73.6% | 87.5% | **72.8%** | 73.5% | 52.5% | 65.5% | 86.5% | 86.0% |
| No Transformer | Teacher 2 - Seed 1 | N/A | **72.5%** | 70.4% | 51.5% | 81.3% | 73.2% | 86.2% | **66.6%** | 72.5% | 40.5% | 57.0% | 83.5% | 79.5% |
| No Transformer | Teacher 1 - Seed 2 | N/A | **73.4%** | 78.5% | 48.6% | 80.9% | 73.0% | 85.9% | **70.2%** | 72.5% | 51.0% | 64.0% | 86.5% | 77.0% |
| No Transformer | Teacher 2 - Seed 2 | N/A | **73.1%** | 76.3% | 48.8% | 82.1% | 72.2% | 86.0% | **67.6%** | 69.5% | 38.0% | 58.0% | 87.5% | 85.0% |
| No image augmentation | Teacher 1 - Seed 1 | N/A | **74.1%** | 79.4% | 50.2% | 81.4% | 73.3% | 86.1% | **66.6%** | 77.5% | 44.0% | 69.5% | 81.0% | 61.0% |
| No image augmentation | Teacher 2 - Seed 1 | N/A | **71.4%** | 76.5% | 47.8% | 81.6% | 65.0% | 86.2% | **60.1%** | 67.5% | 32.0% | 44.5% | 86.0% | 70.5% |
| No image augmentation | Teacher 1 - Seed 2 | N/A | **74.1%** | 74.4% | 51.0% | 82.4% | 72.9% | 89.6% | **70.3%** | 78.0% | 52.0% | 61.5% | 90.0% | 70.0% |
| No image augmentation | Teacher 2 - Seed 2 | N/A | **71.6%** | 76.2% | 45.8% | 80.6% | 70.2% | 85.1% | **56.9%** | 63.0% | 26.5% | 48.5% | 79.0% | 67.5% |
| No action delay | Teacher 1 - Seed 1 | N/A | **74.5%** | 79.9% | 50.6% | 83.9% | 72.8% | 85.3% | **71.1%** | 74.5% | 52.5% | 71.5% | 83.5% | 73.5% |
| No action delay | Teacher 2 - Seed 1 | N/A | **72.1%** | 75.2% | 47.8% | 81.3% | 70.7% | 85.3% | **67.2%** | 73.0% | 45.5% | 56.5% | 89.5% | 71.5% |
| No action delay | Teacher 1 - Seed 2 | N/A | **74.0%** | 77.5% | 51.3% | 83.6% | 73.7% | 84.1% | **68.9%** | 75.0% | 43.5% | 69.5% | 80.5% | 76.0% |
| No action delay | Teacher 2 - Seed 2 | N/A | **73.3%** | 74.8% | 52.3% | 82.6% | 71.3% | 85.4% | **67.6%** | 66.5% | 57.5% | 58.5% | 81.5% | 74.0% |
| MSE & no binary gripper | Teacher 1 - Seed 1 | N/A | **72.2%** | 74.9% | 53.6% | 78.4% | 71.3% | 82.8% | **30.8%** | 44.5% | 23.5% | 7.0% | 20.0% | 59.0% |
| MSE & no binary gripper | Teacher 2 - Seed 1 | N/A | **60.0%** | 74.8% | 0.3% | 72.6% | 65.4% | 81.8% | **24.2%** | 36.0% | 0.0% | 0.0% | 24.5% | 60.5% |
| MSE & no binary gripper | Teacher 1 - Seed 2 | N/A | **72.4%** | 73.8% | 55.5% | 78.0% | 71.0% | 83.7% | **24.4%** | 45.0% | 24.0% | 0.5% | 13.5% | 39.0% |
| MSE & no binary gripper | Teacher 2 - Seed 2 | N/A | **59.1%** | 73.18% | 0.2% | 75.0% | 66.3% | 81.0% | **23.2%** | 27.0% | 0.0% | 0.5% | 24.5% | 64.0% |
| No binary gripper | Teacher 1 - Seed 1 | N/A | **74.4%** | 77.2% | 53.1% | 81.4% | 73.4% | 87.0% | **43.6%** | 61.5% | 36.5% | 0.0% | 61.5% | 58.5% |
| No binary gripper | Teacher 2 - Seed 1 | N/A | **57.8%** | 72.2% | 1.0% | 68.0% | 66.8% | 80.9% | **8.8%** | 10.0% | 0.0% | 0.5% | 19.0% | 14.5% |
| No binary gripper | Teacher 1 - Seed 2 | N/A | **74.5%** | 76.5% | 52.5% | 82.2% | 74.3% | 86.9% | **16.4%** | 55.5% | 12.5% | 0.0% | 8.0% | 6.0% |
| No binary gripper | Teacher 2 - Seed 2 | N/A | **57.7%** | 70.2% | 1.0% | 69.9% | 67.5% | 80.0% | **13.9%** | 24.0% | 0.0% | 0.5% | 29.0% | 16.0% |
| *Skill Generalization* | | | | | | | | | | | | | | |
| IIL-s2r | Teacher 1 - Seed 1 | 65.7% | **58.9%** | 38.1% | 38.5% | 44.7% | 82.5% | 90.5% | **54.8%** | 30.0% | 42.0% | 42.5% | 93.0% | 66.5% |
| IIL-s2r | Teacher 1 - Seed 2 | 64.4% | **59.2%** | 35.2% | 42.3% | 48.0% | 80.8% | 89.8% | **49.6%** | 22.4% | 44.5% | 29.5% | 91.5% | 60.0% |
| IIL-s2r | Teacher 2 - Seed 1 | 64.8% | **53.8%** | 15.7% | 40.9% | 40.7% | 81.4% | 90.5% | **54.6%** | 27.0% | 41.0% | 39.5% | 89.0% | 76.5% |
| IIL-s2r | Teacher 2 - Seed 2 | 63.8% | **52.1%** | 12.5% | 40.8% | 36.4% | 80.0% | 91.0% | **48.4%** | 20.5% | 30.5% | 35.0% | 91.0% | 65.0% |
| No Transformer | Teacher 1 - Seed 1 | 61.5% | **52.7%** | 22.9% | 38.3% | 34.4% | 79.2% | 88.6% | **43.6%** | 30.0% | 22.5% | 14.0% | 82.5% | 69.0% |
| No Transformer | Teacher 1 - Seed 2 | 57.0% | **45.1%** | 11.7% | 35.7% | 16.7% | 75.5% | 85.7% | **44.4%** | 24.5% | 28.5% | 16.0% | 79.0% | 74.0% |
| No Transformer | Teacher 2 - Seed 1 | 64.4% | **54.2%** | 15.3% | 41.1% | 40.9% | 81.2% | 92.3% | **47.8%** | 31.5% | 36.0% | 17.5% | 88.5% | 65.5% |
| No Transformer | Teacher 2 - Seed 2 | 63.6% | **53.2%** | 17.8% | 39.4% | 38.3% | 80.6% | 90.3% | **45.7%** | 24.5% | 26.5% | 27.5% | 87.5% | 62.5% |
| No object parameters | Teacher 1 - Seed 1 | 65.0% | **50.8%** | 29.9% | 15.0% | 38.3% | 82.6% | 88.0% | **32.1%** | 23.0% | 18.5% | 24.0% | 36.0% | 59.0% |
| No object parameters | Teacher 1 - Seed 2 | 65.0% | **58.0%** | 29.9% | 41.0% | 49.9% | 81.9% | 87.4% | **29.3%** | 24.5% | 21.0% | 21.5% | 27.5% | 52.0% |
| No object parameters | Teacher 2 - Seed 1 | 61.7% | **52.2%** | 24.0% | 33.3% | 39.0% | 78.5% | 86.0% | **53.5%** | 29.5% | 42.0% | 29.0% | 87.5% | 79.5% |
| No object parameters | Teacher 2 - Seed 2 | 62.3% | **53.8%** | 24.0% | 38.7% | 38.6% | 80.6% | 86.9% | **49.6%** | 30.0% | 33.5% | 30.0% | 84.0% | 70.5% |

**Table S12: Sim-to-Real Transfer Success.** Ablations of the components of the sim-to-real policy. This table gives a full account of all evaluations for the equivalent Table 2 in the main paper. We execute the stochastic and deterministic policies in simulation and on the robots, respectively, unless otherwise specified.

| Method | Run | Real-Robot Success | | | | | |
|---|---|---|---|---|---|---|---|
| | | Triplet Avg. | Triplet 1 | Triplet 2 | Triplet 3 | Triplet 4 | Triplet 5 |
| Scripted agent | N/A | **51.2%** | 36.3% | 23.0% | 34.4% | 84.9% | 77.6% |
| *Skill Mastery* | | | | | | | |
| BC (scripted agent data) | Seed 1 | **50.8%** | 25.5% | 22.5% | 35.0% | 85.5% | 85.5% |
| BC (scripted agent data) | Seed 2 | **55.9%** | 43.5% | 31.0% | 41.5% | 83.5% | 80.0% |
| CRR (scripted agent data) | Seed 1 | **40.4%** | 17.0% | 11.4% | 41.5% | 66.0% | 66.0% |
| CRR (scripted agent data) | Seed 2 | **46.4%** | 24.0% | 44.5% | 35.0% | 62.0% | 66.5% |
| —Sim-to-real agent for test triplets data | N/A | **69.6%** | 76.4% | 52.7% | 60.4% | 86.5% | 72.0% |
| BC-IMP (sim-to-real agent data) | Seed 1 | **75.1%** | 76.0% | 59.5% | 70.0% | 90.5% | 79.5% |
| BC-IMP (sim-to-real agent data) | Seed 2 | **74.1%** | 75.0% | 62.0% | 71.5% | 85.0% | 77.0% |
| CRR-IMP (sim-to-real agent data) | Seed 1 | **81.0%** | 88.0% | 66.5% | 74.0% | 88.0% | 88.5% |
| CRR-IMP (sim-to-real agent data) | Seed 2 | **82.1%** | 86.5% | 70.0% | 76.5% | 88.5% | 89.0% |
| *Skill Generalization* | | | | | | | |
| —Suboptimal agent for training set data | N/A | **32.6%** | 21.5% | 16.5% | 17.0% | 60.0% | 48.0% |
| BC-IMP (suboptimal agent data) | Seed 1 | **48.2%** | 23.0% | 33.0% | 37.0% | 82.0% | 66.0% |
| BC-IMP (suboptimal agent data) | Seed 2 | **49.8%** | 23.0% | 45.5% | 41.5% | 73.0% | 66.0% |
| CRR-IMP (suboptimal agent data) | Seed 1 | **54.6%** | 27.5% | 42.0% | 41.0% | 79.5% | 83.0% |
| CRR-IMP (suboptimal agent data) | Seed 2 | **56.5%** | 35.0% | 40.5% | 43.0% | 83.5% | 80.5% |

**Table S13: Real-Robot Success.** Different approaches for solving our RGB-stacking tasks in the real world. This table gives a full account of all evaluations for the equivalent Table 3 in the main paper, except for IIL-s2r which are given in Table S12.

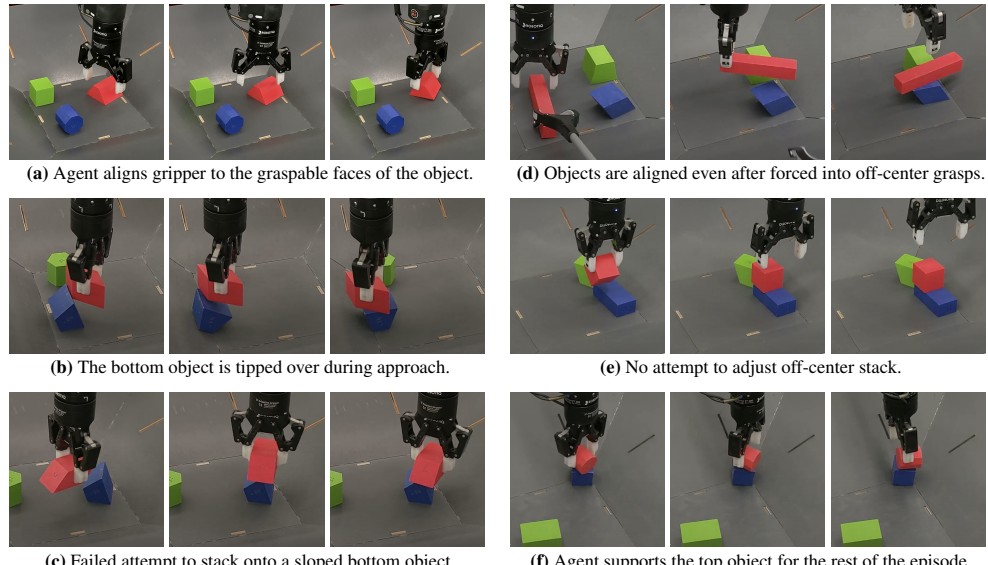

**(a)** Agent aligns gripper to the graspable faces of the object.

**(d)** Objects are aligned even after forced into off-center grasps.

**(b)** The bottom object is tipped over during approach.

**(e)** No attempt to adjust off-center stack.

**(c)** Failed attempt to stack onto a sloped bottom object.

**(f)** Agent supports the top object for the rest of the episode.

**Figure S17:** Agent behaviour during challenging situations.

bottom object's orientation is unfavourable, it will not rotate it to achieve this (Figure S17(c)), and instead try to naively stack the objects.

**Triplet 3.** The main challenge in this triplet lies in the asymmetry of the top object, and needing to balance it onto the bottom one in such a way that their centroids are aligned. Even if disturbed into an off-center grasp, the agent still aligns both objects precisely (Figure S17(d)).

**Triplet 4.** Considered the easiest triplet, the main challenge is to align the centroids, even when the top object can be placed far off-center on the larger bottom object. Perhaps surprisingly, while the agent usually places the object in the center, it shows no attempts to recover from occasional off-center stacks (Figure S17(e)).

**Triplet 5.** Due to the rounded cross-section of the top object, there is a high risk of it rolling off after stacking. As a testament to this, the agent will often stay close to the object after stacking, sometimes (but not always) even supporting it with the gripper until the end of the episode when the bottom object is on a slope and a stack thus otherwise impossible (Figure S17(f)).

## F   Additional Related Work

Our work deals with real-world vision-based stacking with a diverse set of objects and a learned policy. We therefore did not discuss, in the main text, prior work on e.g. stacking from extracted features or in simulation. For completeness, we discuss such works here.

Furrer et al [5] is a very interesting paper on pick-and-place strategies with classical robotics methods that we have now included in our main text discussion. We should highlight here that it deals with only 6 specific stones that offer wide support and have high friction - in contrast to the 152 objects with diverse geometry we propose in our benchmark. To the best of our understanding, their method would neither be able to handle Triplet 2 (flipping the bottom object if needed), nor generalize to unseen objects, as is the case for our "Skill Generalization" task. We should also note that the evaluation on that work was done on a total of 11 episodes (or a maximum of 33 possible stacks in that setup) - in contrast to the more than 54,000 episodes we have evaluated our various choices with.

Duan et al [60] deals with cube stacking in simulation only with policies that have access to task demonstrations, the state information of each cube and no visual input. Later work by Li et al [61] in the same environment shows that reinforcement learning without demonstrations can learn to stack the cubes from state. Our stacking task is also related to other manipulation tasks, such as dry stacking [62, 63] where rocks of irregular shapes must be stacked to form a wall, but these methods

do not address. However, while these methods deal with high-level planning and goal understanding, our benchmark task requires dealing with low-level contact dynamics and perception to make real object stacking possible with strategies that emerge from RL training in simulation.

Noseworthy et al [64] also deals with high-level planning for stacking cubes, this time in the real world. The challenge in this work is that each cube was created with a slightly different center of mass, requiring precise stacks. However, this is not a vision-based task: each cube is also AR-tagged and therefore privileged information about each cube are known during evaluation. Similarly, Macias et al [65] use a combination of binary markers on objects as well as high level planning to perform pick and place stacking.

Some vision-based methods have addressed a related but distinct problem of predicting stack stability [66, 67, 68, 69]. Lerer et al [67] deal with intuition around physics and identifying whether a tower block will collapse, and even the trajectory the blocks will take, focusing primarily on simulation experiments. Similarly, Hamrick et al [69] deal with identifying stability of simulated block towers, with the aid of Graph Neural Networks and without using vision. They also attempt to address which parts of the tower to "glue" in order to fix an unstable tower. Groth et al [68] classify structures as stable or unstable from visual inputs and learn "stackability affordances" for objects of various shapes. However, this line of work does not address the dynamic aspects of manipulation, as there is no physical robot interacting with the objects.

# G   Additional Supplementary Material

In this appendix we attempted to provide as much detail as we could regarding our benchmark, our environments, and our implementation and experimental details. As part of the material that supplement this paper, we also include a video that shows our agents in action[13], as well as designs and instructions for recreating the real robotic cell[14] and the STL files for the RGB-Objects in Figure S2[15].

Here is a list of what is included in the supplementary material as part of this submission:

- Video that supplements the main text.
- BOM (Bill of materials, list of things and quantity) to build:
    - Cell
    - Basket
- Building instructions:
    - Cell
    - Basket
    - Wiring diagram

We will further release, post-submission under an Apache Licence, the following:

- 3D Assembly drawing to complete / integrate assembly instruction
- 3D models and Manufacturing drawing for all the parts (cell and Basket) that need to be:
    - Machined
    - 3D printed
    - Laser Cut
- All STL files for the training set, held-out set, and the specific triplets, in separate folders.
- A version of the simulated environment.

---

[13]Video: https://dpmd.ai/robotics-stacking-YT.
[14]Real cell documentation: https://github.com/deepmind/rgb_stacking/tree/main/real_cell_documentation.
[15]RGB-objects:    https://github.com/deepmind/dm_robotics/tree/main/py/manipulation/props/rgb_objects.

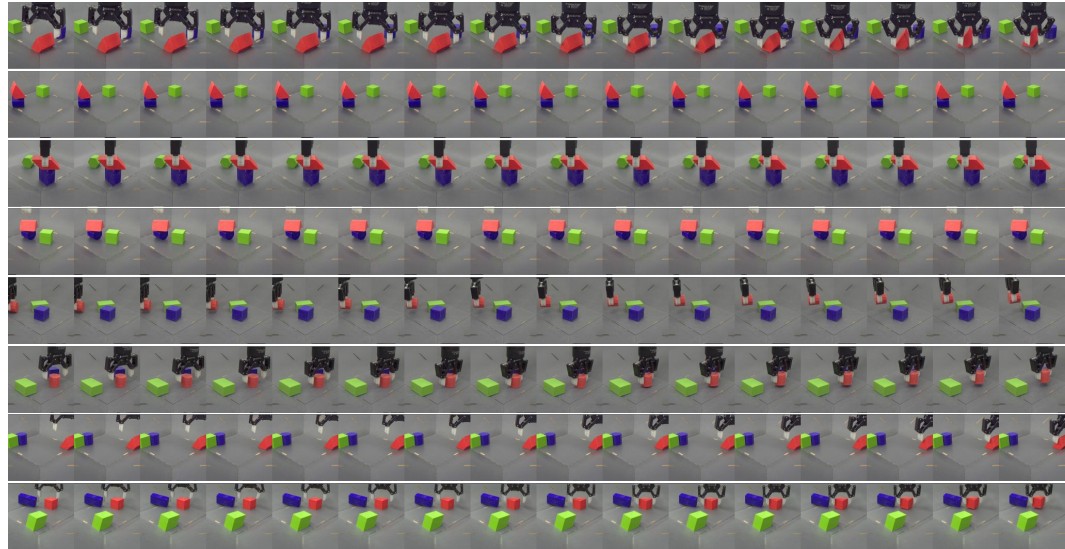

**(a)** Real-world image sequences without image augmentations.

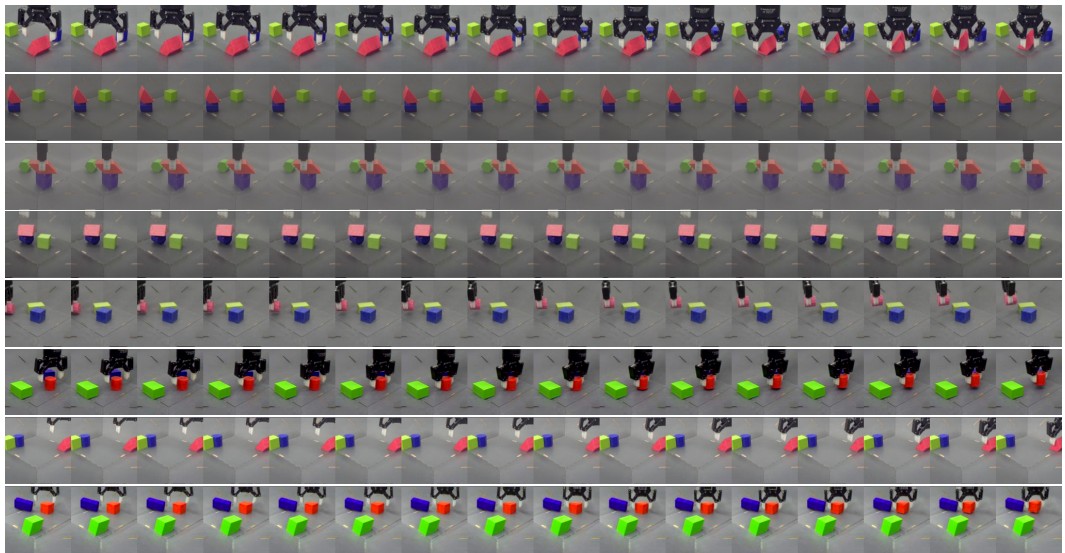

**(b)** Real-world image sequences with image augmentations.

**Figure S18:** Real-world image sequences without and with the image augmentations listed in Table S7. Note that we randomly sample different color and translation perturbations for each sequence in a batch, but we use the same random perturbations for all the images within a sequence.

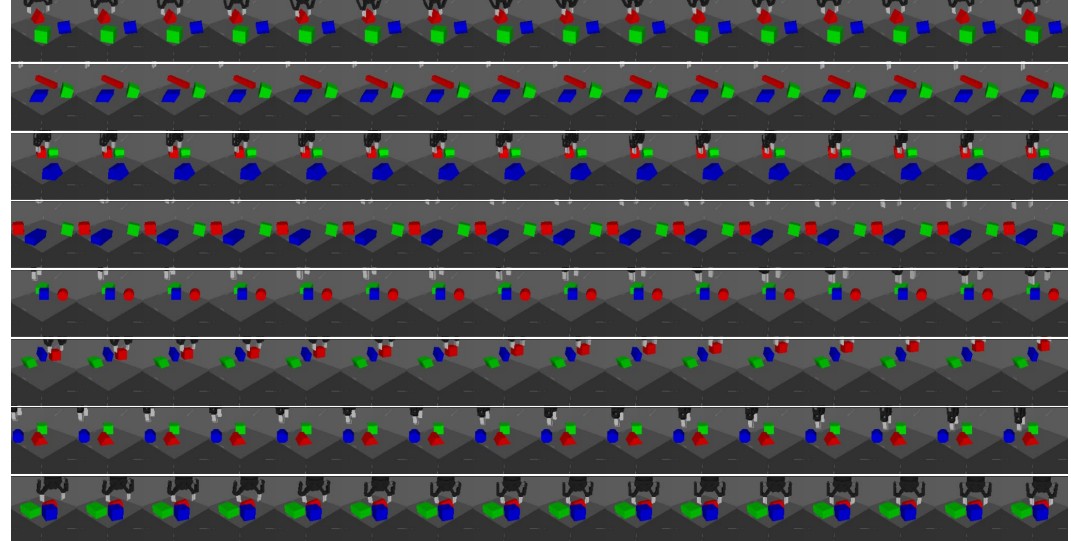

(a) Image sequences from the simulation without image augmentations.

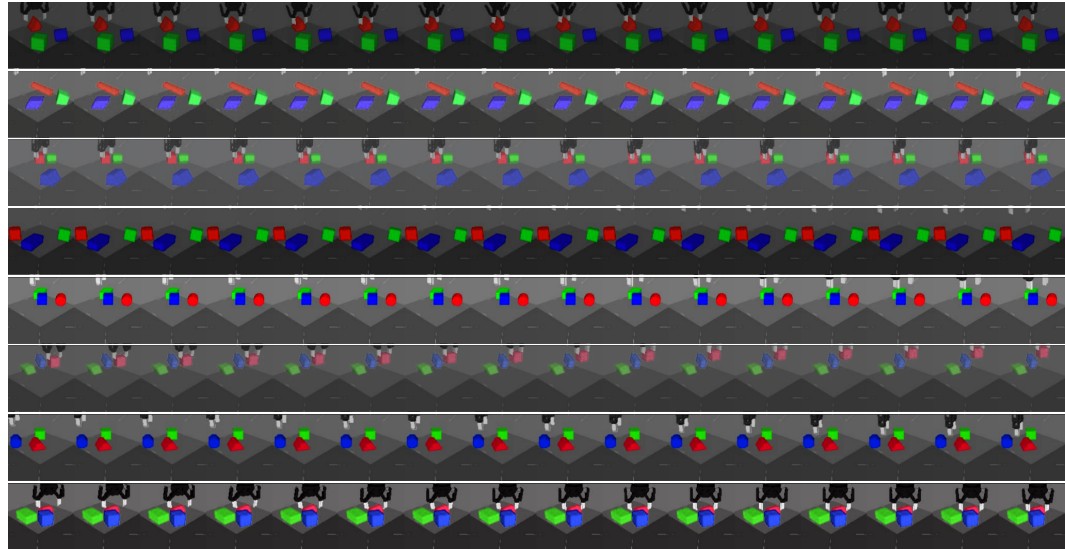

(b) Image sequences from the simulation with image augmentations.

**Figure S19:** Image sequences from simulation without and with the image augmentations listed in Table S7. Note that we randomly sample different color and translation perturbations for each sequence in a batch, but we use the same random perturbations for all the images within a sequence. Although in our experiments we always use image augmentation in combination with domain randomization, here we show example sequences without domain randomization for visualization clarity. See Figure S20 for examples with domain randomization.

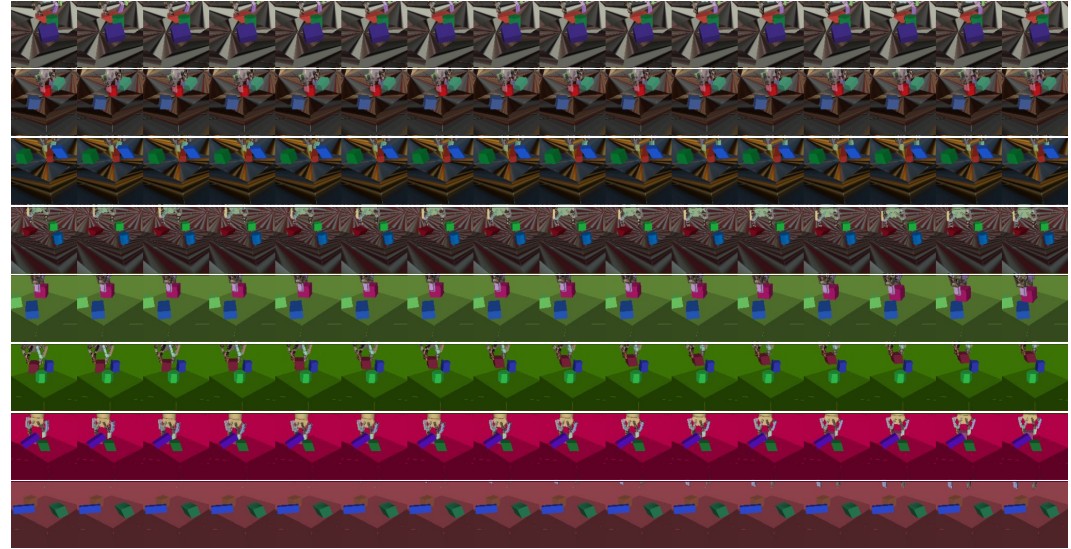

(a) Image sequences from the domain-randomized simulation without image augmentations.

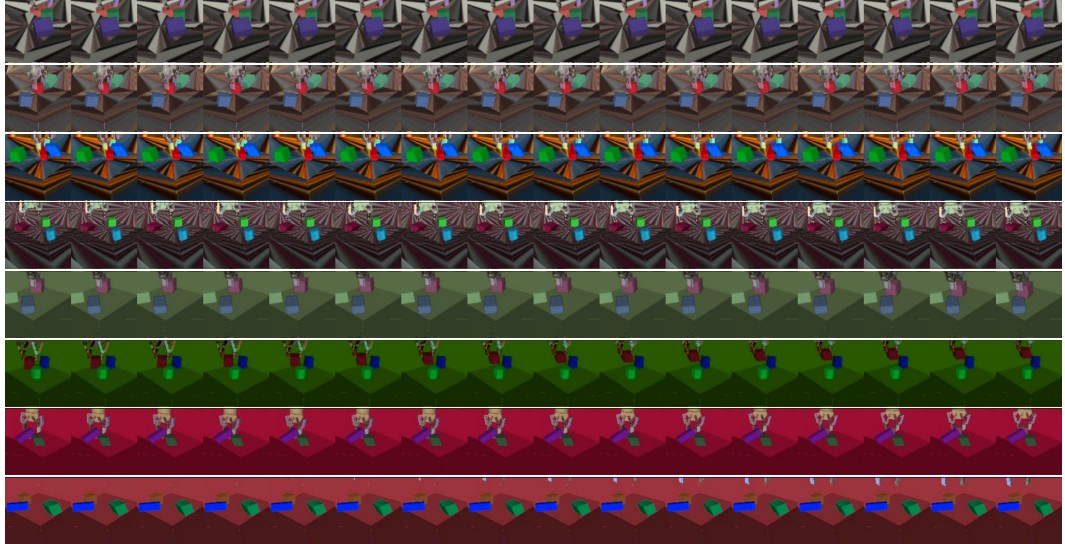

(b) Image sequences from the domain-randomized simulation with image augmentations.

**Figure S20:** Image sequences from the domain-randomized simulation without and with the image augmentations listed in Table S7. Note that we randomly sample different color and translation perturbations for each sequence in a batch, but we use the same random perturbations for all the images within a sequence. See Figure S19 for examples without domain randomization.

