# OpenReview forum: "Beyond Pick-and-Place: Tackling Robotic Stacking of Diverse Shapes"
_robot-learning.org/CoRL/2021/Conference — CoRL2021 Poster_

### Official Review · Reviewer_gHPD · 2021-07-23

**Originality:** Good
**Technical Quality:** Good
**Clarity Of Presentation:** Good
**Impact:** 3

**Recommendation:**

Weak Accept: I recommend accepting the paper, but will not argue for my recommendation if the majority of other reviewers have a different opinion.

**Summary:**

This paper proposes a method to deal with the problem of robotic stacking with objects of different geometry by training a vision-based reinforcement learning, and a benchmark for stacking geometric objects is presented. They successfully achieve skill mastery of fixed combinations of objects and also assess skill generalization across them. To achieve sim-to-real transfer, interactive imitation learning is used to distill the state-based policy into a vision-based policy, and the final policy receives relatively high accuracy in the real world. Besides, lots of real-world evaluations including ablation study are provided and discussed, showing what matters in this task.

**Issues:**

List issues to be addressed during the period of author response and revision. The authors have the opportunity to submit a revised manuscript during the period.


**Reviewer Expertise:**

Good: General knowledge of the area

**Strengths And Weaknesses:**

Strength
1. This paper proposes a benchmark for the interesting but challenging problem of stacking objects with complex geometry.
2. High accuracy is achieved in real-world experiments, and a large ablation study is provided for sim-to-real transfer and imitation of a simulation-trained RL policy.

Weakness
1. The expert policies from state features in simulation are conditioned on parameter ‘y’, which means the object triplet number in skill mastery or object deformation number in skill generalization. However, I think it’s not necessary to add this parameter because the information is contained in the states in simulation or the visual observations in the real world. It might limit the capacity of generalization in this work.


**Summary Of Recommendation:**

This paper proposes a benchmark for the problem of stacking objects with complex geometry, and achieves good performance in real-world experiments. However, I think the limited variety of triplets during training limits the contribution of this work because there exist endless object combinations in the real world, which might reduce the policy’s capacity for generalization.

---

> ### Author Response · Authors · 2021-08-26
> **Response to Reviewer gHPD**
>
> Thank you very much for your review. We were glad to read that you found our benchmark interesting and challenging.
>
> We would like to explain the use of conditioning the state-based policies on parameter ‘y’ a bit further.
> We should start by highlighting that this information is *not* available in the real world and so it is also not included as part of the input for the vision-based policies (IIL, BC, or CRR). Although it is not strictly necessary to add this parameter for our state-based policies either (see Table 2-No object parameters), the information is actually not contained in the states in simulation, as that only includes the position of the center of mass and the object pose for each object. This information is thus agnostic to object identity or shape parameters. By providing this additional input we enable our state-based agents to come up with a diverse set of stacking policies tailored to various shapes.
>
> We agree that the variety of triplets during training (~1 million possible triplets) is limited compared to the endless object combinations in the real world, but it is still a big step above what has been attempted in prior work, and a pretty difficult, still unsolved problem. We have also added two additional videos to our supplementary material with random unseen everyday objects that we found in the lab, qualitatively testing the limits of generalization with our setup. One shows *all* the successes (“diverse_object_successful.mp4”) and the other shows *all* the failures (“diverse_objects_unsuccessful.mp4”) from our attempts, to provide a clear picture. *(Currently, the submission of our revised supplementary zip file is pending due to problems with the submission process. We will upload our revised file once the problem is fixed.)*

---

### Official Review · Reviewer_SRSs · 2021-07-23

**Originality:** Good
**Technical Quality:** Good
**Clarity Of Presentation:** Fair
**Impact:** 3

**Recommendation:**

Weak Accept: I recommend accepting the paper, but will not argue for my recommendation if the majority of other reviewers have a different opinion.

**Summary:**

This paper studies robotic stacking of diverse rigid objects. It introduces a benchmark for robotic stacking and proposes a framework that learns a policy to solve the tasks in simulation and transfer the learned model to the real world. The proposed framework achieves a reasonable task success rate in both simulation and the real world.

**Issues:**

I would suggest the authors re-think their contributions in this paper and make major revisions to Sec. 1 and Sec. 4. I would also encourage the authors to compare with state-of-the-art sim-to-real transfer learning approaches to justify the significance of their technical contributions.

**Reviewer Expertise:**

Very good: Comprehensive knowledge of the area

**Strengths And Weaknesses:**

Strengths:

- In this paper, the authors design and implement a simulation environment for stacking rigid objects. A parametric set of objects is proposed to create diverse challenges for the stacking problem. This simulation environment could incentivize interesting future research on table-top robotic manipulation if the authors will release the codebase.

-   The proposed sim-to-real transfer algorithm seems to be effective. The model achieves a reasonable success rate of stacking diverse objects in simulation and the real world.

Weaknesses:

- The contributions of this paper are confusing. Although the title claims "Beyond Pick-and-Place", the object stacking task studied in this paper is indeed composed of picking and placing actions. In addition, while stacking diverse objects have been studied in prior work (e.g. Furrer et al ICRA 2017), the title focuses on the application without highlighting any technical novelties proposed in this paper. The authors claim that "We show that it is possible to learn a single vision-based policy that can handle multiple combinations of objects and can demonstrate a plethora of stacking strategies, emergent from RL training" However, this has already been widely studied in the community and demonstrated in prior work such as Duan et al. NeurIPS 2017.

- The introduction section of the paper fails to explain the importance of the problem and does not mention any previous works or their limitations. The first two paragraphs mainly talk about the task design and the experiment setups.

- The term and the formulation of "sim-to-real policy" in Sec. 4.2 and 4.3 are misleading. In Sec. 4.2, the authors train the policy in simulation and expect it to be directly executed in the real world. So there is anything related to sim-to-real transfer about this policy $\pi_{\theta}^{s2r}$ itself. It would be better to call it just student policy or something else.

- Although it seems that the proposed sim-to-real transfer approach is one of the main contributions in this paper, the paper lacks a thorough comparison with state-of-the-art sim-to-real transfer approaches in the experiment. Without such comparisons, it is hard to justify the significance of this paper.

- Many important papers are missing in the related work section, including but not limited to:
    - Furrer et al. ICRA 2017
    - Macias et al. ICRA 2014
    - Noseworthy et al. RSS 2021
    - Duan et al. NeurIPS 2017

- (minor) The listed keywords include CoRL which seems to be a mistake.


**Summary Of Recommendation:**

The paper studies an interesting robotic manipulation problem and proposes a valid technical solution. However, the contributions of this paper are not clearly described or justified given its current form and the lacking of comparisons with prior work. Therefore, I believe this work requires another iteration before being ready for publication.

---

> ### Author Response · Authors · 2021-08-26
> **Response to Reviewer SRSs**
>
> We would like to thank you for your review and your suggestions for improvement which we hope to have addressed to your satisfaction. We were happy to read that you believe our simulation environment could incentivize future research and we can commit that we will indeed make it publicly available by mid-October 2021.
>
> We understand your concerns and we have attempted to address them all. We have made major revisions to Sections 1, 2, and 4 in order to make our contributions even clearer in the context of related work, and highlight the importance of the problem. In the meantime, please let us know if you have any additional questions we might be able to answer. We also respond here explicitly to the concerns raised.
>
> **Title and Contributions**
>
> We understand the confusion regarding our title and contributions. To the best of our knowledge, our work is the first to demonstrate vision-based stacking of non-cuboid objects in the real world at scale and with such level of success: prior works on cuboids saw success rates of 20-60% in contrast to our 82% on a diverse set of challenging object shapes. What differentiates our work are: 1) the large set of objects with diverse geometric shapes and the complex object dynamics these elicit, and 2) the large-scale evaluations with which we validate our claims. The novelty and significance of our work indeed lies in our empirical study of this stacking application, our benchmark results highlighting difficulties in the generalization setting, and our level of success not previously seen in literature. We believe we have not claimed that we propose a novel method, sim-to-real or otherwise, and we have made this clearer in our revisions.
>
> Our title accordingly focuses on our application, rather than the methodological novelty. The reason we have included the phrase “Beyond Pick-and-Place” in our title is because we do have evidence that strategies qualitatively different and more difficult than simple pick-and-place movements are required to solve certain configurations of the task —most notably Triplet 2 where the bottom object often needs to be flipped for a successful stack. We observe that such behavior indeed emerges during RL training of our general policies and successfully transfers to the real robots.
>
> **Suggested Related Work**
>
> Our work deals with real-world vision-based stacking with a diverse set of objects and a learned policy. We therefore did not discuss, in our submission, prior work on e.g. stacking from extracted features or in simulation. We will now include such a discussion in an additional section of our supplement by the end of the week. We have also revised Sects. 1, 2, 4 in a major way in order to highlight what separates our work from prior art and to make our contributions even clearer.
>
> Furrer et al, ICRA 2017 is a very interesting paper on pick-and-place strategies with classical robotics methods that we have now included in our main text discussion. We should highlight here that it deals with only 6 specific stones that offer wide support and have high friction - in contrast to the 152 objects with diverse geometry we propose in our benchmark. To the best of our understanding, their method would neither be able to handle Triplet 2 (flipping the bottom object if needed), nor generalize to unseen objects, as is the case for our “Skill Generalization” task. We should also note that the evaluation on that work was done on a total of 11 episodes (or a maximum of 33 possible stacks in that setup) - in contrast to the more than 54,000 episodes we have evaluated our various choices with.
>
> Duan et al, NeurIPS 2017 is an excellent paper that deals with cube stacking in simulation only with policies that have access to the state information of each cube and no visual input. While they deal with high-level planning and goal understanding, we attempt to deal with low-level contact dynamics and perception to make real stacking possible with strategies that emerge from RL training in simulation, which is one of the three components in our proposed framework.
>
> **Sim-to-Real Policy**
>
> We understand the confusion re: our presentation of our methods and we believe we have now addressed it. In our original submission, we called our vision-based student policy a “sim-to-real policy” because it is trained with a number of choices and components specifically designed to improve transfer to the real world, e.g. domain randomization [19,23] and image augmentation. We have rename the policy to $\pi^{vis}_\theta$, and also revised Sect. 4 to make this point clearer.

---

> > ### Comment · Reviewer_SRSs · 2021-09-04
> > **Response to the Authors**
> >
> > I appreciate the authors' efforts in improving the paper and clarifying the contributions. The paper is now more ready for publication. Therefore, I have raised my rating to Weak Accept.

---

### Official Review · Reviewer_JZ71 · 2021-07-24

**Originality:** Very Good
**Technical Quality:** Very Good
**Clarity Of Presentation:** Excellent
**Impact:** 4

**Recommendation:**

Strong Accept: I recommend accepting the paper and will argue for my recommendation even if other reviewers hold a different opinion.

**Summary:**

*This work developed a novel benchmark in the context of stacking manipulation. The benchmark is novel and challenging. The setup cost is reasonable and details have been provided for the ease of reproduction which is valuable to the community.

*It is shown to be possible to learn a single vision-based policy that can handle multiple combinations of objects and can demonstrate a plethora of stacking strategies, emergent from RL training. Zero-shot transfer of these strategies to the real robot has been achieved.

*The ablation studies suggest what matters for sim-to-real transfer and for imitation of a simulation-trained RL policy (that uses privileged state information) with a vision-based general policy. The proposed interactive distillation approach allowed for faster experimentation and iteration.


**Issues:**

In the video 0:44, in the first row, isn’t the triplet5 a failure case? Triplet3’s execution is also not complete.

**Reviewer Expertise:**

Good: General knowledge of the area

**Strengths And Weaknesses:**

Strengths:

*The developed benchmark is novel, challenging and interesting. It involves diverse kinds of objects while requiring different manipulation strategies. The setup details have also been provided for reproduction.

*The paper is well written and easy to follow.

*The design choices are reasonable and has been reflected in the ablation study.

Weaknesses:

There is no obvious weakness found, except for some minor issues explained below.


**Summary Of Recommendation:**

Accept.

---

> ### Author Response · Authors · 2021-08-26
> **Response to Reviewer JZ71**
>
> Thank you very much for your review and suggestions. We are glad to see that you find that our benchmark and findings will be valuable to the community.
> Regarding your concerns: Indeed Triplet 5 seems like a failure case and we have now addressed this for the final video. We have also now further improved our manuscript and have added two additional videos in our supplement that showcase how our agents perform with random everyday objects. *(Currently, the submission of our revised supplementary zip file is pending due to problems with the submission process. We will upload our revised file once the problem is fixed.)*

---

> > ### Comment · Reviewer_JZ71 · 2021-09-03
> > **Thanks**
> >
> > thanks for the updates.

---

### Official Review · Reviewer_6zgt · 2021-07-24

**Originality:** Good
**Technical Quality:** Very Good
**Clarity Of Presentation:** Very Good
**Impact:** 3

**Recommendation:**

Weak Accept: I recommend accepting the paper, but will not argue for my recommendation if the majority of other reviewers have a different opinion.

**Summary:**

This paper presents a new benchmark of completing pick-and-place tasks using a robot manipulator. In comparison to existing pick-and-place tasks that deal with simple-shape objects, this paper described a parametric way of generating more than a hundred of blocks of diverse shapes that were 3D printed. The agent first learned a policy from simulation using RL methods, where the domain is fully observable (object positions, etc). Then imitation learning methods were used to generate a vision-based policy, where domain randomization was enabled for alleviate the sim-to-real issue. Finally, the policy learned from simulation was improved for real-world deployment.

**Issues:**

See above.

**Reviewer Expertise:**

Good: General knowledge of the area

**Strengths And Weaknesses:**

The 3D-printed parametric block generation is a strength of this paper, and has the potential of providing a strong benchmark to the community. So the first contribution listed on page 1 is solid: "We present and release a benchmark for stacking that features
a diverse set of geometric objects and tens of thousands of possible stacks." It's also good that the paper used state of the art methods for learning policies in simulation and in the real world. It will be better if the objects shown in Figure 1 (d) can go beyond toy blocks that have been widely used in robot manipulation research.

The downside is that the paper does not make it clear if any of the used algorithms is new from the literature. As a result, the reviewer is not convinced that the second contribution is valid: "We show that it is possible to learn a single vision-based policy that can handle multiple combinations of objects and can demonstrate a plethora of stacking strategies."  There are many existing methods for vision-based manipulation, and there are competitions on this robot manipulation. The reviewer respectfully believes that there is rich literature on learning vision-based policies for manipulating irregular-shape objects. The indication that no existing work support vision-based handling of multiple objects is questionable. There's a similar concern on contribution (d), which is a claim about sim-to-real policy learning. Improving sim-to-real policies using real-world data is not new.

Contribution (c) on ablation study makes sense. However, since those are all mature methods from the literature, one probably won't say it's a major contribution to the community.

**Summary Of Recommendation:**

Overall, the reviewer didn't observe anything incorrect from the paper, and the paper was nicely generated. However, the paper mostly demonstrated that existing methods can be applied to vision-based policy learning for manipulation tasks on a new 3D-printed set of objects. The contribution is solid but not significant.

---

> ### Author Response · Authors · 2021-08-26
> **Response to Reviewer 6zgt**
>
> Thank you for the review and suggestions for improving the paper. We have now incorporated them in a new revision. We would like to clarify that in this paper we propose a new benchmark and an empirical study of said benchmark. We have not claimed that improving sim-to-real policies using real-world data, or the algorithms we use in different steps are novel. We have now made this clearer in Sect 1 to address your concern. We do believe, however, that our findings are important and useful to the community. We discuss a framework we have used with which we were able to achieve stacking of the objects in the benchmark, using vision and sim-to-real transfer. This framework separates training in 3 phases which not only significantly increased our development and research velocity, but made tackling these tasks possible at all. These are 1) RL training (MPO) in simulation from state, 2) interactive imitation learning with domain randomization to obtain a vision-based policy for zero-shot transfer, and 3) one-step improvement with offline RL (CRR) and data collected by the sim-to-real vision-based policy. We demonstrate that using a combination of existing methods for these phases can lead to strong results.
>
> Re: our second contribution, it was not our intention to indicate that there is no existing work that supports vision-based handling of multiple objects. What we show in this work is that a single, learned, vision-based policy is able to demonstrate different strategies specifically for *stacking* diverse objects. Although there is plenty of literature on manipulating real diverse objects, we are not aware of any work that has learned vision-based policies for stacking diverse objects or for other tasks that are not semi-planar on real robots. Stacking objects with diverse geometries is a significant step forward because it requires inter-object interaction that involves complex contact dynamics. It is also the first logical step after grasping and lifting, on the way, for example, to constructing more complex structures with multiple objects.
>
> Although we do deal with 3D-printed toy shapes, this choice allows us to study the problem in a principled way because we can control the affordance of these objects. We should highlight that prior work primarily deals with toy cuboids (limited affordance variety) that can be solved with simpler pick-and-place strategies devised by classical robotics algorithms (e.g. our scripted agent baseline). Our work is a big step away from that. The best example of why our objects pose an important challenge is our evaluation Triplet 2 (where one of the objects needs to be flipped). Additionally, in the spirit of standardization and reproducibility we wanted to introduce a set of objects that can be 3D-printed while still being very challenging to stack. Widely available commercial 3D-printing, however, does pose certain limitations on the textures and material that can be used. We have also now included, as part of our supplementary material, two videos that show how our agents can (and cannot) generalize to unseen real objects that are not toy shapes. *(Currently, the submission of our revised supplementary zip file is pending due to problems with the submission process. We will upload our revised file once the problem is fixed.)*
>
> Finally, we would like to politely disagree re: the significance of our ablation study. While a couple of our ablations may seem dull (e.g. the effect of image augmentation on top of domain randomization for sim-to-real transfer), many of our findings were non-obvious to us and we believe will be interesting to practitioners. One example for this is the fact that for our “Skill Generalization” task, using the object parameters with a non-Transformer policy, or a Transformer policy without the object parameters as part of the input were not enough to improve performance. (See Table 2 - Skill Generalization). These findings, which we believe generalize beyond the specific setup we have considered here, were evaluated with multiple runs of the same settings and many thousands of episodes on the real robots, which took more than a month of continuous operations to complete.

---

### Author Response · Authors · 2021-08-28
**Supplementary Material Revision Now Available**

Previously, due to a problem with OpenReview, we could not upload our revised supplementary material zip file. The OpenReview issue is now fixed and we have uploaded our revised supplement, which includes:
* an additional section (Sect F) in our revised Appendix on "Additional Related Work" on stacking from states or solely in simulation.
* two additional videos of our agents attempting to stack unseen everyday objects found in our lab.
* a revision of our main video to address minor required fixes.

Thank you again for your time in reviewing our work.

---

### Meta-Review · Area_Chair_An1x · 2021-08-13

**Recommendation:** Accept (Poster)
**Confidence:** 4

**Metareview:**

This paper presents a new benchmark for stacking manipulation. Reviewers are overall positive about the paper however there are some concerns:
1. Relationship to prior work
2. Lack of comparison to state of art

Please respond to reviewer SRSs concerns.

==
Post Rebuttal

The authors were able to address the concerns of the reviewers. AC believes this is a strong paper which should be accepted.

---

> ### Author Response · Authors · 2021-08-26
> **Summary of Changes and Response to the Meta Review**
>
> **Overview**
>
> We are happy to see that reviewers were overall positive about our paper and the benchmark we are proposing. We believe we have addressed the concerns of all our reviewers as well as we possibly could. Most notably we have revised Sections 1, 2, and 4 in order to make our contributions and their relationship to prior work clearer, as suggested by Reviewers 6zgt and SRSs. We have also updated our submitted main video with minor fixes suggested by Reviewer JZ71, and have provided two additional videos qualitatively testing the generality of our agents with various everyday objects and in a number of different settings unseen during training (eg different colors, more than 3 objects in the basket etc) in order to provide our reviewers, and especially Reviewer gHPD, with more insight into our agents’ capacity for generalization. These episodes were not cherry-picked: the successes and failures are separated into two videos “diverse_object_successful.mp4” and “diverse_objects_unsuccessful.mp4”.
> *(Currently, the submission of our revised supplementary zip file is pending due to problems with the submission process. We will upload our revised file once the problem is fixed.)*
>
>
> Specifically for the concerns raised in this metareview:
>
> **Re: 1. Relationship to prior work**
>
> Our work deals with real-world vision-based stacking with a diverse set of objects and a learned policy. We therefore did not discuss, in our submission, prior work on stacking from extracted features or in simulation. We will now include such a discussion in an additional section of our supplement by the end of the week. As per reviewer SRSs’s request, we have revised and improved Sects. 1, 2, 4 in order to highlight what separates our work from prior art and to make our contributions even clearer.
>
> **Re: 2. Lack of comparison to state of art.**
>
> We should highlight that our paper’s contribution is not about introducing a novel algorithm but about proposing a novel, challenging benchmark, and the “application of robot learning in robot manipulation”, which is one of the areas the CoRL CfP is explicitly soliciting contributions on. As such, comparison to eg more complex sim-to-real strategies, as requested by our reviewers, is beyond the scope of this work.
>
> Specifically, our contributions are the following: we describe a framework which is able to learn to stack objects with complex geometries in the real world, using vision and sim-to-real transfer. The learning process in our proposed framework draws from several state of the art methods in the literature, including domain randomization [19,23], dataset aggregation [15], and critic-regularized regression [24,25], and these choices are ablated in Tables 1, 2, 3, and S12 (requiring more than 54,000 stacking attempts on real robots). For example, in Table 1 we find that using data from the student policy, as is done in [15], leads to higher performance than only training on data from the teacher policy. We have also proposed a new, carefully designed benchmark of 152 objects with which to evaluate robotic stacking both in simulation and the real world and provide, in our supplement (Sect A.2), quantitative and qualitative analysis of some of the challenges it poses. Finally, we have included all relevant information required to reproduce our real-world setup as part of our supplementary material, and are publicly releasing, by mid-October 2021, a version of our sim stacking environment, the models for our 152 RGB objects, and instructions for 3D printing them. All the above, we believe, are valuable resources to practitioners and other researchers in the field of learning robot manipulation.

---

> > ### Public Comment · ~Andrew_Hundt1 · 2021-10-27
> > **Relevant Related Work**
> >
> > Hello, this benchmark is certainly quite extensive at a time when research work that creates integrated systems can be significantly undervalued. Even so, I wanted to inquire about some claimed contributions with respect to prior work. I’ll also mention upfront that I’m an author on [1, 3], two of the seven papers I reference below:
> >
> > > (a) We present and release a benchmark for stacking that features a diverse set of geometric objects and tens of thousands of possible stacks.
> > > (b) We show that it is possible to learn a vision-based policy that can stack multiple combinations of objects and can demonstrate a variety of stacking strategies for non-cuboid objects, emergent from RL training.
> >
> > [1] completes sim-to-real raw-image RGBD vision-based long-horizon manipulation tasks including cube stacking and row-making, with a simulated budget of 20,000 actions, and a real budget of 0 actions. [1] also discusses how cube stacking requires learning about stack stability, in contrast to paragraph 2 of your introduction. [1] also has important limitations, some of which are addressed by this submission.
> >
> > [1] can be evaluated for stacking with 8 objects of varied shapes and many colors with a command line parameter change. No [code](https://github.com/jhu-lcsr/good_robot) or data changes are required.
> >
> > [2] uses a stacking benchmark like [1] and makes a couple of sim-only improvements on long-horizon tasks. [3] has a real robot offline dataset benchmark for blocks stacking with physical constraints and varied soft distractor objects. [4] is a paper with varied shape decluttering which serves as the baseline benchmark for [1] and [2].
> >
> > > (c) We present the framework we used to obtain our results without the need for human demonstrations.
> >
> > [1] is an RL framework that successfully obtains results on stacking and row tasks without human demonstrations, with very successful sim-to-real transfer for cubes, but the sim-to-real component does not succeed for all evaluated tasks. [1] also has RGBD input and a very different control scheme, with scripted position control actions, rather than the velocity control described by this submission.
> >
> > Side note: There are significant potential downsides to large scale benchmarks that are worthy of consideration, for example, a large scale benchmark might constitute a costly prerequisite that prevents scientifically valid results by lower-resourced organizations from being accepted for publication [5, 6, 7]. Might the authors consider straightforward adjustments that maintain benchmark validity and increase accessibility for lower resourced research groups for the future? For example, 3D printing and velocity control increase manufacturing and training costs, respectively.
> >
> > While some of the prior work this submission already described is applicable to prior work I cite here, not all of it is. I’d also like to recognize that, as the authors mentioned, the CoRL call for proposals specifically calls for “Applications of robot learning in robot manipulation”.
> >
> > Finally, [1, 3, 4] were published well before the June 18, 2021 CoRL submission deadline, but [2, 5, 6, 7] are from around or after the submission date and/or are preprint only, and this is the reason I specified a side note.
> >
> > Thank you for your time and consideration of these matters.
> >
> > ---
> >
> > [1] A. Hundt, B. Killeen, N. Greene, H. Wu, H. Kwon, C. Paxton, and G. D. Hager, “‘‘Good Robot!”: Efficient Reinforcement Learning for Multi-Step Visual Tasks with Sim to Real Transfer,” IEEE Robotics and Automation Letters (RA-L), vol. 5, no. 4, pp. 6724–6731, 2020. DOI: https://doi.org/10.1109/LRA.2020.3015448. [Online]. Available: https://arxiv.org/abs/1909.11730.
> >
> > [2] Sulabh Kumra, Shirin Josh, Ferat Sahin "Learning Robotic Manipulation Tasks through Visual Planning" ArXiV e-prints https://arxiv.org/abs/2103.01434
> >
> > [3] A. Hundt, V. Jain, C. Lin, C. Paxton and G. D. Hager, "The CoSTAR Block Stacking Dataset: Learning with Workspace Constraints," IROS, 2019, pp. 1797-1804, doi: https://doi.org/10.1109/IROS40897.2019.8967784.
> >
> > [4] A. Zeng, S. Song, S. Welker, J. Lee, A. Rodriguez and T. Funkhouser, "Learning Synergies Between Pushing and Grasping with Self-Supervised Deep Reinforcement Learning," IROS, 2018, doi: https://doi.org/10.1109/IROS.2018.8593986.
> >
> > [5] A. Birhane, P. Kalluri, D. Card, W. Agnew, R. Dotan, and M. Bao, The values encoded in machine learning research, 2021. arXiv: 2106.15590 [cs.LG]. [Online]. Available:  https://arxiv.org/abs/2106.15590
> >
> > [6] M. Brandão, “Normative roboticists: The visions and values of technical robotics papers,” in RO-MAN, 2021. DOI: https://doi.org/10.1109/RO-MAN50785.2021.9515504. [Online]. Available: https://www.martimbrandao.com/papers/Brandao2021-roman-visions.pdf.
> >
> > [7] J. S. O. Ceron and P. S. Castro, “Revisiting rainbow: Promoting more insightful and inclusive deep reinforcement learning research,” in ICML, PMLR, Jul. 2021. [Online]. Available: https://proceedings.mlr.press/v139/ceron21a.html.

---

### Decision · Program_Chairs · 2021-09-13

**Decision:**

Accept (Poster)

**Comment:**

This paper presents a new benchmark for stacking manipulation. Reviewers are overall positive about the paper however there are some concerns:
1. Relationship to prior work
2. Lack of comparison to state of art

Please respond to reviewer SRSs concerns.

==
Post Rebuttal

The authors were able to address the concerns of the reviewers. AC believes this is a strong paper which should be accepted.